# Single turnover transient state kinetics reveals processive protein unfolding catalyzed by *Escherichia coli* ClpB

**Jaskamaljot Kaur Banwait, Liana Islam, Aaron L Lucius\***

Department of Chemistry, University of Alabama at Birmingham, Birmingham, United States

**Abstract** *Escherichia coli* ClpB and *Saccharomyces cerevisiae* Hsp104 are AAA+ motor proteins essential for proteome maintenance and thermal tolerance. ClpB and Hsp104 have been proposed to extract a polypeptide from an aggregate and processively translocate the chain through the axial channel of its hexameric ring structure. However, the mechanism of translocation and if this reaction is processive remains disputed. We reported that Hsp104 and ClpB are non-processive on unfolded model substrates. Others have reported that ClpB is able to processively translocate a mechanically unfolded polypeptide chain at rates over 240 amino acids (aa) per second. Here, we report the development of a single turnover stopped-flow fluorescence strategy that reports on processive protein unfolding catalyzed by ClpB. We show that when translocation catalyzed by ClpB is challenged by stably folded protein structure, the motor enzymatically unfolds the substrate at a rate of ~0.9 aa s$^{-1}$ with a kinetic step-size of ~60 amino acids at sub-saturating [ATP]. We reconcile the apparent controversy by defining enzyme catalyzed protein unfolding and translocation as two distinct reactions with different mechanisms of action. We propose a model where slow unfolding followed by fast translocation represents an important mechanistic feature that allows the motor to rapidly translocate up to the next folded region or rapidly dissociate if no additional fold is encountered.

**\*For correspondence:**
allucius@uab.edu

## eLife assessment

This **valuable** study presents the development of a single turnover stopped-flow fluorescence experiment to study the kinetics of substrate unfolding and translocation by the bacterial ClpB disaggregase. Using non-physiological nucleotides to bypass the physiological regulation mechanism of ClpB, the authors **convincingly** show that the ClpB disaggregase is a processive motor with a slow unfolding step preceding rapid translocation. The results of this analysis are of value for future mechanistic studies on energy-dependent unfolding, degradation, and disaggregation molecular machines.

## Introduction

ATPases associated with diverse cellular activities (AAA+) play essential roles in cell physiology, such as proteome maintenance, membrane fusion, DNA replication, repair and recombination, RNA processing, chromatin remodeling, and organelle biogenesis (*Ogura and Wilkinson, 2001*; *Snider et al., 2008*; *Shorter and Houry, 2018*; *Seraphim and Houry, 2020*; *Neuwald et al., 1999*). Across domains of life, representative AAA+ molecular motors are essential for proteome maintenance (*Erzberger and Berger, 2006*). *Escherichia coli* ClpB and *Saccharomyces cerevisiae* Hsp104 are AAA+ protein disaggregases that, in collaboration with co-chaperones, resolve protein aggregates that form

during heat shock or stress (*Doyle et al., 2007a*; *Glover and Lindquist, 1998*; *Goloubinoff et al., 1999*; *Motohashi et al., 1999*; *Shorter and Lindquist, 2008*; *Zolkiewski, 1999*).

ClpB, in collaboration with the co-chaperones, DnaK, DnaJ, and GrpE (KJE) couple the energy from ATP binding and hydrolysis to disruption of protein aggregates (*Doyle et al., 2007a*; *Goloubinoff et al., 1999*; *Motohashi et al., 1999*; *Zolkiewski, 1999*). However, the molecular mechanisms of protein disaggregation and collaboration with co-chaperones are not fully understood. ClpB is proposed to couple ATP binding and hydrolysis to processive rounds of protein unfolding and translocates the newly extracted polypeptide chain through the axial channel of the hexameric ring structure of the motor. Once passed through the axial channel, in the unfolded state, the polypeptide can refold or interact with co-chaperones that would aid in protein refolding.

Two strategies have been developed to isolate the activity of ClpB without the need for the co-chaperones, KJE. First, several 'hyperactive' ClpB variants have been identified where single-point mutations in the motor relieve the need for the co-chaperones, for example ClpB(Y503D) (*Oguchi et al., 2012*). The other strategy is using a mixture of ATP and the slowly hydrolysable ATP analog, ATPγS. Wickner and co-workers discovered that a 1:1 mixture of ATP:ATPγS could activate ClpB in a manner that alleviated the need to include the co-chaperones, KJE, thereby simplifying *in vitro* studies (*Doyle et al., 2007b*).

In those *in vitro* experiments the polypeptide substrate being processed enters and leaves the reaction with structural changes but no change in molecular weight because ClpB and Hsp104 are not proteases. This fact has limited our knowledge on the mechanisms of ClpB and Hsp104 catalyzed protein unfolding and translocation. This limitation contrasts with the homologous AAA+ molecular motors, ClpA and ClpX (*Olivares et al., 2018*; *Duran et al., 2017*), which processively unfold and translocate a polypeptide into the proteolytic barrel of the associated ClpP. Thus, the product of the protein unfolding and translocation reaction catalyzed by ClpA or ClpX is covalently modified by the associated protease, ClpP. In turn, proteolytic fragments are used as a signal to report on translocation.

To interrogate ClpB and Hsp104 catalyzed polypeptide translocation we developed and applied a rapid mixing stopped-flow fluorescence method (*Li et al., 2015b*; *Durie et al., 2019*). In that work we used unstructured polypeptides so that the time courses would reflect the kinetics of translocation and not protein unfolding. Independent of substrate length we detected only two steps before the enzymes dissociated from the unfolded polypeptides. Those results were interpreted to indicate that both ClpB and Hsp104 exhibit low processivity (*P* between 0.37 and 0.61) during polypeptide translocation of an unfolded chain (*Li et al., 2015b*; *Durie et al., 2019*). However, we could not rule out the possibility that these motors might proceed through two slow steps followed by rapid translocation at a rate that is outside the millisecond temporal resolution of the stopped-flow technique. It is also possible that undetectable rapid translocation is followed by slow rate-limiting dissociation from the end.

Consistent with rapid translocation on an unfolded protein, Avellaneda et al. reported translocation rates for ClpB(Y503D) to be ~240 and 450 aa s$^{-1}$ on mechanically unfolded substrates (*Avellaneda et al., 2020*). However, neither their reported rates of ATP hydrolysis nor any published rates are consistent with this hyper-fast translocation activity (*Lin et al., 2022*). Similarly, Mazal et al. reported ultrafast pore-loop dynamics and correlated this with rapid translocation of the unfolded κ-casein substrate (*Mazal et al., 2019*; *Mazal et al., 2021*). Mazal et al. acknowledged that these domain movements are substantially faster than ATP turnover.

To examine enzyme catalyzed protein unfolding, Wickner and co-workers constructed RepA(1-70)-GFP, which contains the N-terminal 70 amino acids of the phage P1 RepA protein followed by green fluorescent protein (GFP) (*Hoskins et al., 2000*). The N-terminal sequence of RepA serves as a binding site for ClpB. They showed a loss of GFP fluorescence upon exposure of this construct to ClpB in the presence of a 1:1 mixture of ATP:ATPγS (*Doyle et al., 2007b*), which was interpreted to indicate ClpB unfolded GFP. They further showed no loss of fluorescence over a 45-min time frame when only ATP or only ATPγS was provided. It is important to note that loss of GFP fluorescence does not indicate that the substrate was processively translocated after the unfolding event that led to loss of fluorescence.

Our previous examinations of ClpB and Hsp104 were carried out on unstructured polypeptide chains, so we interpret those results to reflect translocation and not protein unfolding (*Li et al., 2015b*; *Durie et al., 2019*). Here, we sought to develop a single turnover transient state kinetics approach

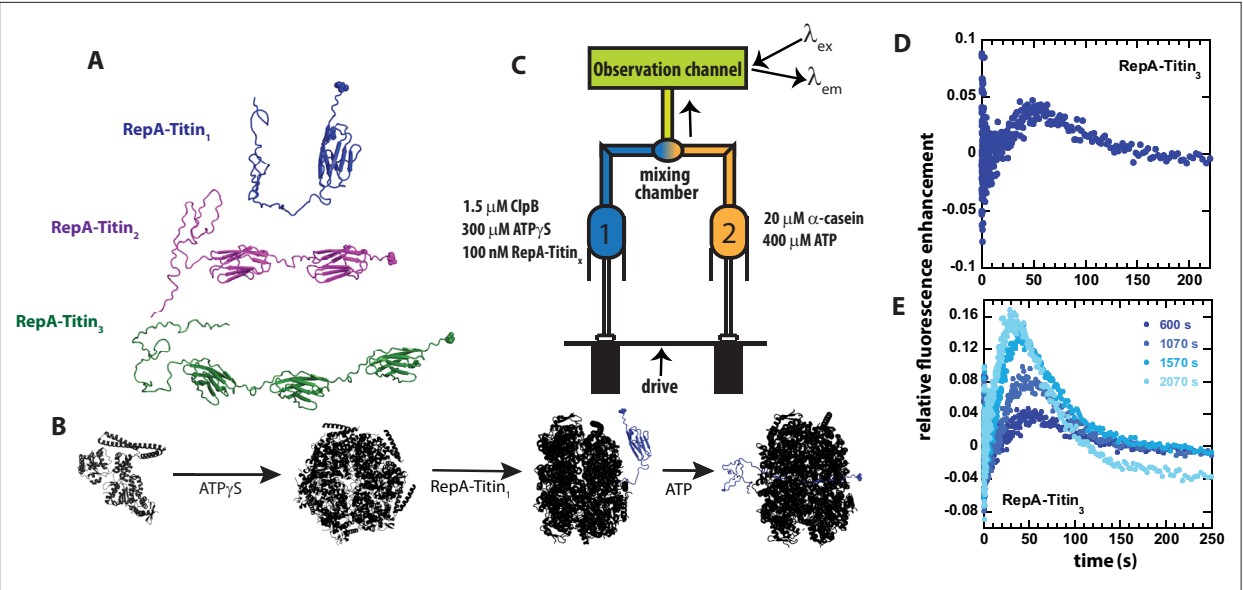

**Figure 1.** Single turnover ClpB catalyzed protein unfolding. (**A**) RepA(1-70)-Titin$_1$ (Blue), RepA(1-70)-Titin$_2$ (purple), and RepA(1-70)-Titin$_3$ (green). Each construct from N- to C-terminus consists of the first 70 amino acids of the Phage P1 RepA protein, a known binding sequence for ClpB followed by tandem repeats of the Titin I27 domain separated by linkers. Each construct contains a single cysteine shown in space-filling at the C-terminus that has been reacted with Alex Fluor (AF)–555. (**B**) Schematic of steps in forming pre-bound complex based on our previous work (*Lin and Lucius, 2015*; *Lin and Lucius, 2016*; *Weaver et al., 2017*; *Li et al., 2015a*). ClpB (black) is assembled into hexameric rings competent for substrate binding by adding ATPγS, illustrated as bound to the RepA-Titin$_1$ substrate in blue, followed by rapid mixing with ATP. As shown, ClpB is expected to unfold the Titin I27 domains and translocate the newly unfolded substrates through the axial channel of the hexameric ring. (**C**) Schematic representation of stopped-flow. Syringe 1 contains the indicated concentrations of ClpB monomer, ATPγS, and RepA-Titin$_X$, where X = 1, 2, 3. Syringe 2 contains 400 μM ATP and 20 μM α-casein to serve as a trap for any free ClpB. The contents of the two syringes are rapidly mixed at a 1:1 mixing ratio and flow into the observation channel where AF555 is excited at $\lambda_{ex}$ = 555 nm and emission is observed at $\lambda_{em}$ > 570 nm. (**D**) Representative time-course collected using strategy in (**C**) using RepA-Titin$_3$ after pre-incubating the sample at 25°C for 600 s. (**E**) Successive experimental time-courses collected as in D. The total time of incubation before collection of the time-course is indicated.

The online version of this article includes the following source data and figure supplement(s) for figure 1:

**Source data 1.** The data points for relative fluorescence enhancement of *Figure 1D, E* are tabulated as a function of time for each incubation period of 600, 1070, 1570, and 2070 s.

**Figure supplement 1.** Testing protein-induced fluorescence enhancement with an unstructured substrate of 50 amino acids.

**Figure supplement 1—source data 1.** The sheet labeled 'Figure 1—figure supplement 1A' consists of data points for emission signal as a function of wavelength shown in *Figure 1—figure supplement 1A*.

**Figure supplement 2.** A zoomed-in version of *Figure 1E* to show the decreasing lag with increasing incubation time of the pre-bound complex.

to test for processive unfolding catalyzed by ClpB. Inspired by the RepA(1-70)-GFP constructs made by Wickner and co-workers (*Doyle et al., 2007b*; *Hoskins et al., 2000*) and the Titin I27 substrates used extensively by the Baker and Sauer groups (*Kenniston et al., 2003*) we constructed RepA-Titin$_X$, where X = 1, 2, or 3, see *Figure 1A*. In these constructs the first 70 amino acids of the RepA protein provide the binding site and are followed by one, two, or three repeats of the Titin I27 domain. Each protein ends with a C-terminal cysteine used for fluorescent labeling by maleimide chemistry.

Using the RepA-Titin$_X$ constructs, we report evidence of sequential and processive unfolding of the tandem repeats of the stably folded Titin I27 domains. Surprisingly, we have also found that ATPγS alone will support processive protein unfolding and translocation of these constructs. Because of this we developed a sequential mixing stopped-flow strategy to separate and quantify the rates of ATPγS- and ATP:ATPγS-driven protein unfolding. Here, we report rates of protein unfolding in the range of 1–4 amino acids (aa) s$^{-1}$ with a kinetic step-size of ~60 aa unfolded between two rate-limiting steps. These rates of protein unfolding are approximately two orders of magnitude slower than the reported translocation rates on unfolded polypeptide chains reported by others (*Avellaneda et al., 2020*; *Mazal et al., 2021*). Thus, we propose that protein unfolding catalyzed by ClpB is rate-limiting and, upon unfolding, translocation on the newly unfolded polypeptides is much faster than protein

unfolding. Our method reveals mechanistic insights into ClpB catalyzed protein unfolding that have been inaccessible by other techniques. Importantly, our technology overcomes the barrier of needing covalent modification to detect enzyme catalyzed protein unfolding and translocation by the protein disaggregating machines. The approach can be broadly applied to the many AAA+ motors that have been hypothesized to catalyze protein unfolding and translocation but do not covalently modify the substrate on which they operate.

## Results

### Development of single turnover protein unfolding method

To test for protein unfolding catalyzed by wild type (wt) ClpB, we engineered constructs containing the N-terminal 70 amino acids of the RepA protein *Doyle et al., 2007b*; *Durie et al., 2018* followed by tandem repeats of the Titin I27 domain. Each construct contains a single cysteine residue at the C-terminus labeled with Alexa Fluor (AF) 555-maleimide, see *Figure 1A*. The rationale for these constructs is that we can preassemble ClpB into the biologically active hexamers in the presence of ATPγS (*Lin and Lucius, 2015*; *Lin and Lucius, 2016*; *Weaver et al., 2017*) and bind the hexamer to the N-terminal RepA sequence (*Li et al., 2015a*), see *Figure 1B* for schematic of assembly and binding steps.

The pre-bound complex is loaded into Syringe 1 of the stopped-flow fluorometer. In the other syringe is ATP and excess α-casein. The α-casein serves as a trap for any unbound ClpB, see *Figure 1C* for schematic of stopped-flow setup. The contents of the two syringes are rapidly mixed within 2 ms, AF555 is excited at $\lambda_{ex}$ = 555 nm, and emission is observed at $\lambda_{em}$ > 570 nm.

After mixing, if ClpB processively unfolds the Titin I27 domain and translocates N to C, then, upon arrival of ClpB at the AF555, we anticipate protein-induced fluorescence enhancement (PIFE) (*Hwang et al., 2011*), which has been reported to occur with AF555 (*Rashid et al., 2019*). In addition, PIFE was tested using an unstructured 50 aa polypeptide labeled with AF555. When ClpB binds this unstructured 50-mer, PIFE is observed due to the close proximity of AF555 to ClpB, see *Figure 1—figure supplement 1* (*Li et al., 2015b*; *Li et al., 2015a*).

Experiments were performed by stopped-flow mixing of 1.5 µM ClpB, 300 µM ATPγS, and 100 nM RepA-Titin$_3$ with 20 µM α-casein and 400 µM ATP, see *Figure 1C* for mixing schematic. Upon mixing, fluorescence emission from AF555 was monitored as a function of time, see *Figure 1D*. The time-course exhibits a lag phase followed by fluorescence enhancement and gradual loss of signal. The lag is interpreted to indicate the time over which ClpB is unfolding the Titin I27 domains and translocating the newly unfolded polypeptide chain. The fluorescence enhancement is interpreted to indicate arrival of the motor at the C-terminus and induction of PIFE followed by loss of signal due to dissociation.

In these stopped-flow experiments, one fills the contents of the two syringes illustrated in *Figure 1C* and collects multiple time-courses using the same sample upon successive mixing events. The only difference from mixing event to mixing event is that the sample will have aged for the time it took to collect the previous time-course/courses. Thus, the time-courses are expected to overlay between mixing events if the contents of the two syringes are at chemical and thermal equilibrium. Thermal equilibrium is achieved by maintaining contact with a temperature-controlled water bath set to 25 °C and sufficient incubation time. Chemical equilibrium is judged by comparing time-courses collected in succession.

Here, we observed systematic deviation in the collected time-courses from mixing event to mixing event, see *Figure 1E* and a zoomed-in version of *Figure 1E* shown in *Figure 1—figure supplement 2*. The first collected time-course (dark blue) is labeled 600 s as the sample was first allowed to preincubate for 10 min to achieve thermal equilibrium before collection, see *Figure 1D, E*. Select, subsequent time-courses are shown, and the total pre-incubation time is indicated, see *Figure 1E* and *Figure 1—figure supplement 2*. For each successive collection, the lag time is getting shorter, the emergence of a peak occurs earlier in time, and the magnitude of the peak is increasing. The latter is most consistent with binding equilibrium not being fully achieved. But the former two observations suggest that the longer the samples are incubated the less time it takes before PIFE after mixing with ATP. This phenomenon was also observed with RepA-Titin$_1$ and RepA-Titin$_2$, data not shown. These observations suggest that ClpB is moving closer to the fluorophore during the pre-incubation time.

We hypothesized that the variability observed in *Figure 1E* and *Figure 1—figure supplement 2* was due to slow ATPγS hydrolysis being coupled to protein unfolding and translocation catalyzed by

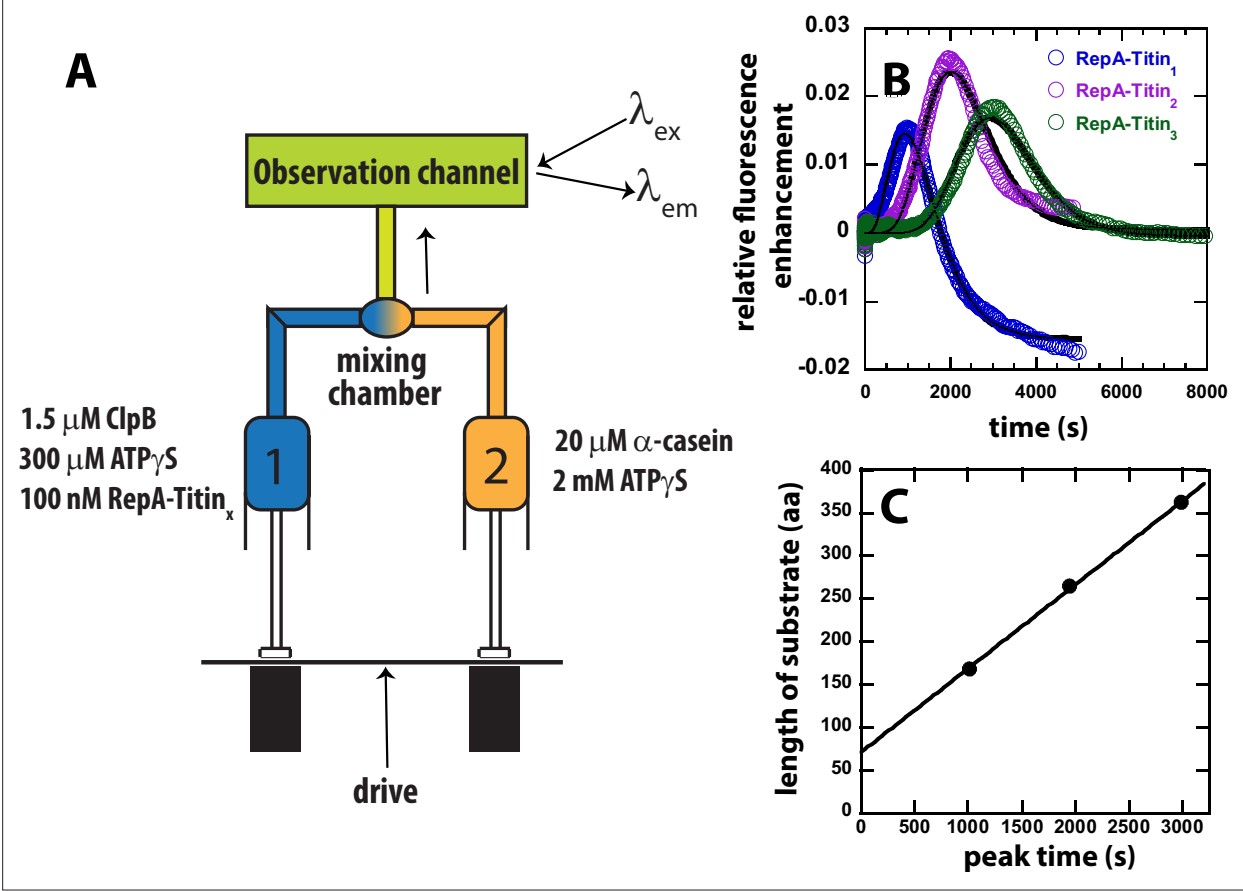

**Figure 2.** Test for ATPγS-driven protein unfolding by ClpB. (**A**) Mixing strategy as in **Figure 1C** but with ATP replaced with 2 mM ATPγS in Syringe 2. (**B**) Time-courses collected using RepA-Titin$_1$ (blue), RepA-Titin$_2$ (purple), and RepA-Titin$_3$ (green) plotted as relative fluorescence enhancement vs. time. The solid black line represents the best-fit line from fitting to Scheme 1, **Figure 4A**. The fitting parameters obtained are the unfolding rate constant, $k_U = (0.0042 \pm 0.0003)$ s$^{-1}$, and the kinetic step-size, $m = (26 \pm 5)$ aa. (**C**) Length of substrate vs. peak time determined from B. The plot was fit to a linear equation to yield a slope = $(0.098 \pm 0.003)$ aa s$^{-1}$ and intercept of $(71 \pm 7)$ aa.

The online version of this article includes the following source data and figure supplement(s) for figure 2:

**Source data 1.** The sheet labeled 'Figure 2B' consists of data points in a tabulated form for each RepA-Titin$_x$ substrate, as shown in **Figure 2B**.

**Figure supplement 1.** Post-mix assay to investigate ATPγS driven protein unfolding catalyzed by ClpB.

**Figure supplement 1—source data 1.** The sheet consists of data points in a tabulated form for each RepA-Titin$_x$ substrate, as shown in **Figure 2— figure supplement 1**. The error bars for each relative fluorescence enhancement data point are tabulated here as standard deviation.

ClpB. Consequently, during the time between mixing events some pre-unfolding and pre-translocation from the N-terminal binding site to the C-terminal fluorophore had already occurred in the syringe.

To test for ATPγS-driven unfolding and translocation, we loaded ClpB pre-bound to RepA-Titin$_1$ in the presence of 300 µM ATPγS into Syringe 1 of the stopped-flow. In Syringe 2, we loaded 2 mM ATPγS and 20 µM α-casein trap with no hydrolysable ATP, see **Figure 2A**. The rationale for this design is that, even though we anticipate pre-translocation due to the presence of 300 µM ATPγS, we expect translocation to be accelerated upon mixing and increasing the total concentration of ATPγS.

In the absence of hydrolysable ATP but in the presence of only ATPγS, the time-courses exhibit a distinct lag, followed by a fluorescence enhancement, and a slow loss in signal, see **Figure 2B**. Consistent with translocation from the N-terminal RepA sequence to the C-terminal fluorophore both the lag time and the time of appearance of the peak is observed to increase for each addition of another Titin I27 domain, see **Figure 2B**. This indicates that ClpB proceeds through an increasing number of rate-limiting steps for each increase in substrate length.

If the observed peak in the time-courses for each RepA-Titin$_X$ substrate, in **Figure 2B**, represent arrival of ClpB at the C-terminal fluorophore then plotting the total length of the substrate

vs. emergence of the peak, termed 'peak time', should represent a classical kinematics position vs. time plot. *Figure 2C* shows the length of the substrate vs. the peak time from the time-courses in *Figure 2B*. As expected, substrate length vs. peak time exhibits a linear increase with a slope of (0.098 ± 0.003) aa s$^{-1}$ and an intercept of (71 ± 7) aa. From these observations, we hypothesize that the slope represents the rate of unfolding and translocation in the presence of a final mixing concentration of 1.15 mM ATPγS and no ATP. We propose that the intercept represents the number of amino acids pre-unfolded and pre-translocated in the presence of 300 µM ATPγS during the 10-min pre-incubation time.

To further test ATPγS-driven translocation, we performed a series of experiments where only ClpB was loaded into Syringe 1 and, in Syringe 2 was loaded ATPγS and RepA-Titin$_x$, see *Figure 2—figure supplement 1A* for mixing schematic. In this setup, before PIFE can be observed, ClpB must bind ATPγS, assemble into hexamers, bind to the RepA sequence, and initiate protein unfolding and translocation. Importantly, these experiments may not be single turnover because the ClpB is not pre-bound to the substrate and there is no trap for free ClpB, that is no α-casein. Nevertheless, the time-courses do exhibit a lag phase that increases with increasing substrate length, followed by PIFE, and an apparent plateau, see *Figure 2—figure supplement 1B*. As expected, when assembly and binding must occur before protein unfolding and translocation can ensue the time-courses exhibit a longer lag and undetectable dissociation after arrival at the fluorophore, compare time-courses in

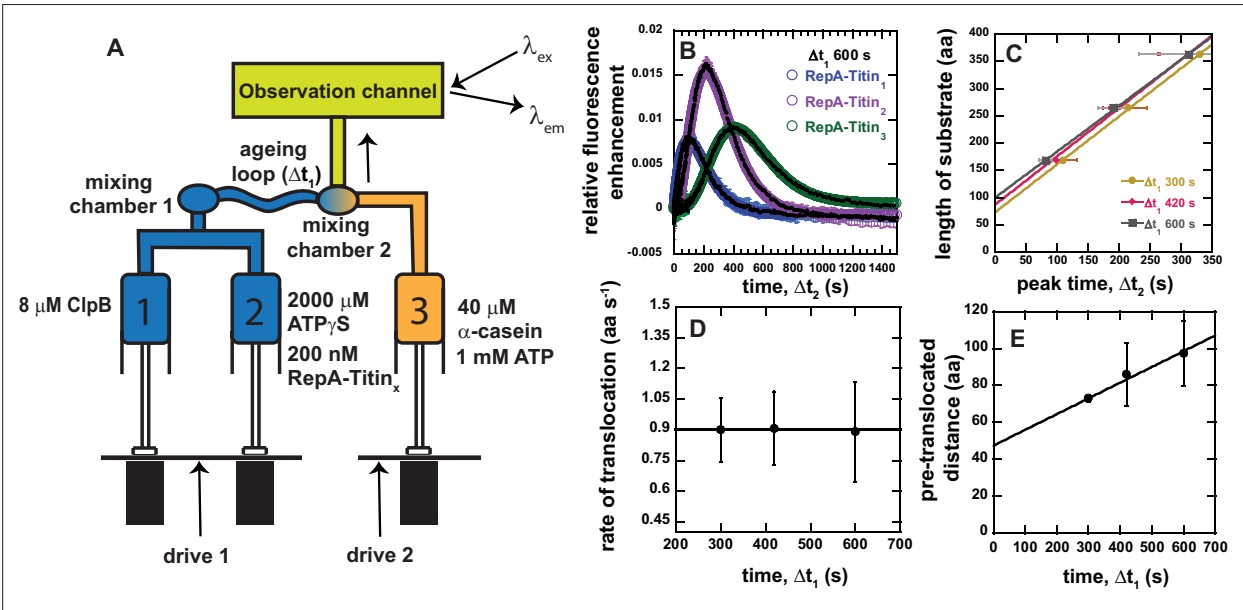

**Figure 3.** Sequential mixing stopped-flow strategy. (**A**) Schematic representation of sequential mixing. Syringe 1 contains 8 µM ClpB monomer, Syringe 2 contains 2 mM ATPγS and 200 nM RepA-Titin$_x$, and Syringe 3 contains 40 µM α-casein and 1 mM ATP. The contents of Syringes 1 and 2 are mixed 1:1 in mixing chamber 1 leading to a concentration of 4 µM ClpB, 1 mM ATPγS, and 100 nM RepA-Titin$_x$. The sample ages for a user-defined amount of time, $\Delta t_1$, followed by rapid mixing with the contents of Syringe 3. The sample flows into the observation channel at a final concentration of 2 µM ClpB, 50 nM RepA-Titin$_x$, 500 µM ATPγS, 500 µM ATP, and 20 µM α-casein. In the observation channel AF555 is excited at $\lambda_{ex}$ = 555 nm and emission is observed at $\lambda_{em}$ > 570 nm. (**B**) Representative time-courses from the average of five or more sequentially collected time-courses for RepA-Titin$_1$(blue), RepA-Titin$_2$(purple), and RepA-Titin$_3$(green) at $\Delta t_1$ = 600 s. The black solid lines represent the best-fit line from fitting to Scheme 1 (see *Figure 4A*). The fitting parameters obtained are $k_U$ = (0.017 ± 0.002) s$^{-1}$ and $m$ = (56.5 ± 0.7) aa. (**C**) Total length of substrate as a function of peak time determined for $\Delta t_1$ = 300, 420, and 600 s. Solid lines represent weighted linear fits yielding (**D**) slope vs. $\Delta t_1$ and (**E**) the intercept vs. $\Delta t_1$. All data points and error bars represent the average and standard deviation determined from three replicates.

The online version of this article includes the following source data and figure supplement(s) for figure 3:

**Source data 1.** The sheet labeled as 'Figure 3B' consists of data points in a tabulated form for each RepA-Titin$_x$ substrate, as shown in *Figure 3B*.

**Figure supplement 1.** Reproducibility using sequential mixing strategy.

**Figure supplement 1—source data 1.** The sheet consists of data points for each consecutive shot in a tabulated form for RepA-Titin$_1$ substrate, as shown in *Figure 3—figure supplement 1*.

**Figure supplement 2.** Cartoon representation showing translocation and unfolding by ClpB on RepA-Titin$_1$ from states I through III: (**I**) Pre-bound complex of ClpB (green) with RepA-Titin$_1$ has dangle distance and contact length unavailable for translocation and unfolding.

*Figure 2—figure supplement 1B* to *Figure 2B*. In sum, all experiments in the presence of only ATPγS as an energy source reveal that ClpB processively unfolds tandem repeats of Titin I27.

## Development of sequential mixing strategy to control and quantify ATPγS-driven protein unfolding and translocation

ClpB is pre-bound to the polypeptide chain to remove the kinetics of ClpB assembly and binding to the protein substrate. This is important as we are seeking to acquire time-courses that are rate-limited only by the kinetics of protein unfolding and/or translocation but not the kinetics of hexamer formation or substrate binding.

Previously, we reported that only ATPγS would support ClpB binding to the polypeptide and the non-hydrolysable nucleoside triphosphate analogs, AMPPNP and AMPPCP could not substitute for ATPγS (*Weaver et al., 2017*). Consequently, we cannot eliminate the ATPγS pre-unfolding/pre-translocation if we seek to have a pre-bound complex. Thus, we devised a sequential mixing strategy to account for and control the pre-unfolding and translocation that occurs in the presence of ATPγS.

Sequential mixing stopped-flow allows for two mixing events, illustrated in *Figure 3A*. In this design we load 8 µM ClpB monomer into Syringe 1 and, in Syringe 2, we load 2 mM ATPγS and 200 nM RepA-Titin$_X$, see *Figure 3A* for mixing schematic. The contents of Syringes 1 and 2 are rapidly mixed and allowed to age, without observation, for a user-defined amount of time, $\Delta t_1$. In this experiment, the $\Delta t_1$ is the amount of time where ClpB will bind ATPγS, assemble into hexamers, bind to RepA-Titin$_X$ and initiate some pre-unfolding and pre-translocation.

After $\Delta t_1$ elapses the sample is rapidly mixed with the contents of the third syringe containing 40 µM α-casein and 1 mM ATP, see *Figure 3A*. After the second mixing event the sample flows into the observation chamber and fluorescence as a function of time, $\Delta t_2$, is monitored. The final concentrations after the two mixing events are 2 µM ClpB monomer, 50 nM RepA-Titin$_X$, 500 µM ATPγS, and 500 µM ATP. The final ratio of ATP to ATPγS was chosen because it has been shown that ClpB exhibits its highest protein disaggregation activity and protein unfolding activity in the presence of a 1:1 mix of ATP:ATPγS. Moreover, the 1:1 mix activates ClpB in the absence of the co-chaperones, KJE (*Doyle et al., 2007b*).

A series of representative time-courses collected with $\Delta t_1$ = 600 s are shown in *Figure 3B*. As expected, we observe a lag and a maximum fluorescence that increases in time with increasing number of Titin I27 domains. Importantly, the representative time-courses in *Figure 3B* for each RepA-Titin$_X$ represent the average of five or more successive time-courses collected using the same sample preparation. Thus, in this design, no variability between mixing events is detected since ClpB and ATPγS have been pre-incubated for precisely the same amount of time, $\Delta t_1$, for each successive mixing event, see *Figure 3—figure supplement 1*.

In all cases, the time-courses collected with RepA-Titin$_2$ exhibit higher peak maximum values compared to RepA-Titin$_1$ and RepA-Titin$_3$. This is because all three substrates are fluorescently labeled to a different labeling efficiency and thus there is a different proportion of labeled to unlabeled substrate in each sample. ClpB will bind to both the labeled and unlabeled substrates and the presence of unlabeled substrate will compete for binding with labeled substrate. RepA-Titin$_2$ exhibits the highest labeling efficiency and thus the lowest concentration of unlabeled substrate to compete for binding. Thus, time-courses collected with RepA-Titin$_2$ always exhibit the highest peak max since the peak max is proportional to extent of binding, see discussion below.

In this sequential mixing design, ClpB will still pre-unfold and pre-translocate the substrate during $\Delta t_1$. To further control and account for ATPγS-driven translocation, we varied $\Delta t_1$ to be 300, 420, and 600 s. The substrate length vs. peak time for each $\Delta t_1$ is plotted and fit to a line, see *Figure 3C*. The slope and intercept for each value of $\Delta t_1$ is plotted in *Figure 3D and E*, respectively.

*Figure 3D* indicates that the rate is independent of $\Delta t_1$ within the time range tested. The average value from the plot in *Figure 3D* is (0.9 ± 0.1) aa s$^{-1}$ and is taken as the rate of unfolding and translocation in the presence of a 1:1 mix of ATP to ATPγS, in this case 500 µM:500 µM.

The intercept values from the fits in *Figure 3C* increase linearly with increasing $\Delta t_1$, see *Figure 3E*. We hypothesize that the intercept represents the number of amino acids pre-unfolded and pre-translocated in the presence of 1 mM ATPγS during $\Delta t_1$, which we term pre-translocated distance. The slope of the pre-translocated distance vs. $\Delta t_1$ yields a value of (0.09 ± 0.06) aa s$^{-1}$, consistent with the rate of translocation determined in the presence of 1.15 mM ATPγS and no ATP, see *Figure 2*.

**Table 1.** Parameters obtained from model-independent analysis on experiments presented in *Figure 3* and *Figure 4—figure supplement 2*.

| Parameters from model-independent analysis | 500 μM [ATPγS] and 500 μM [ATP] | 150 μM [ATPγS] and 500 μM [ATP] |
|---|---|---|
| Rate of translocation with ATPγS and ATP (aa s$^{-1}$) | 0.9 ± 0.1 | 4.3 ± 0.2 |
| Excluded length (aa) | 48 ± 17 | 67 ± 23 |
| Rate of translocation with ATPγS (aa s$^{-1}$) | 0.09 ± 0.06 | 0.05 ± 0.05 |

Error represent the standard deviation determined from three replicates.

Interestingly, the weighted linear fit in *Figure 3E* yields an intercept of (48 ± 17) amino acids when extrapolating to $\Delta t_1 = 0$, see *Table 1*. We interpret this number to represent the average number of amino acids that are not involved in unfolding and translocation, which we will term the excluded length. The excluded length may represent the number of amino acids that are in contact with the motor but not part of the lattice to be translocated, the number of amino acids dangling outside of the hexameric ring relative to the direction of translocation, or some combination of the two, see *Figure 3—figure supplement 2* for schematic representation of these various distances.

### Global fitting of length-dependent time-courses

Scheme 1, in *Figure 4A*, represents an *n*-step sequential mechanism that we propose to describe the experimental time-courses collected at each value of $\Delta t_1$, see *Figure 3B* for $\Delta t_1 = 600$ s. Scheme

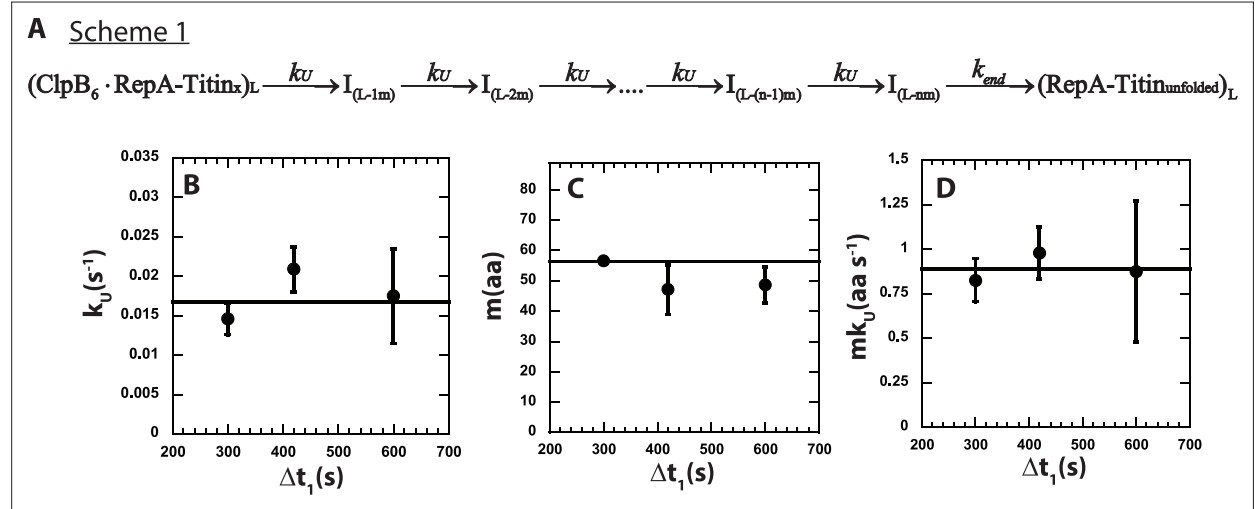

**Figure 4.** *n*-Step sequential mechanism for 500:500 μM ATP:ATPγS. (**A**) Proposed kinetic scheme for ClpB catalyzed protein unfolding and translocation on RepA-Titin$_x$ substrates. Parameters (**B**) $k_U$, (**C**) $m$, and (**D**) $mk_U$ obtained from fitting to Scheme 1 at each $\Delta t_1$ are shown in solid black circles. The black solid line represents the best weighted fit line to a linear equation with zero slope to yield the average value of the unfolding rate constant, $k_U = (0.017 \pm 0.002)$ s$^{-1}$, kinetic step-size, $m = (56.5 \pm 0.7)$ aa, and overall rate, $mk_U = (0.89 \pm 0.09)$ aa s$^{-1}$.

The online version of this article includes the following source data and figure supplement(s) for figure 4:

**Source data 1.** The sheet labeled '*Figure 4B*' consists of data points for the unfolding rate constant, $k_U$ as a function of $\Delta t_1$, represented in *Figure 4B*.

**Figure supplement 1.** Determination of number of steps, *n*, from fitting the time-courses represented in *Figure 3B* to Scheme 1.

**Figure supplement 1—source data 1.** The sheet consists of data points for *Figure 4—figure supplement 1*.

**Figure supplement 2.** ClpB catalyzed protein unfolding at approx. 3:1 ATP:ATPγS.

**Figure supplement 2—source data 1.** The sheet labeled as 'Figure A' consists of data points in a tabulated form for each RepA-Titin$_x$ substrate, as shown in *Figure 4—figure supplement 2A*.

**Figure supplement 3.** Parameters from fitting to Scheme 1 for approx. 3:1 ATP: ATPγS.

**Figure supplement 3—source data 1.** The sheet labeled 'Figure A' consists of data points for the unfolding rate constant, $k_U$ as a function of $\Delta t_1$, represented in *Figure 4—figure supplement 3A*.

1 (see *Figure 4A*) shows ClpB pre-bound to a RepA-Titin$_X$ of substrate length, *L*. Upon mixing with ATP the motor proceeds through *n* number of unfolding steps with rate constant $k_U$. Each intermediate is denoted as $I_{(L\text{-}im)}$, where *L* represents the total number of amino acids in the substrate, *i* is the number of steps taken to arrive at a given intermediate, *I*, and *m* represents the average number of amino acids unfolded between two rate-limiting steps, that is the kinetic step-size. The rate constant, $k_{end}$, represents dissociation from the C-terminal end of the unfolded polypeptide chain denoted as (RepA-Titin$_{unfolded}$)$_L$.

The observed total number of steps, *n*, is hypothesized to be proportional to the substrate length divided by the kinetic step-size as *n* = *L*/*m*. To test this, the three time-courses were simultaneously fit using Scheme 1 (see *Figure 4A* and *Equation 5*) with $k_U$ constrained to be the same for all three time-courses, that is global, and the number of steps, *n*, and $k_{end}$ local to each time-course. In this analysis, we ascribe signal change to the last intermediate, $I_{(L\text{-}nm)}$, and the unfolded product, (RepA-Titin$_{unfolded}$), see Materials and methods.

The determined number of steps as a function of substrate length is plotted in *Figure 4—figure supplement 1A*. The *n* vs. *L* plot was fit to a line with a slope of (0.016 ± 0.002) steps aa$^{-1}$ and an intercept of (−1.48 ± 0.42) steps. As seen in *Figure 4—figure supplement 1A*, the line of best fit exhibits an *x*-intercept of ~93 amino acids. We interpret the intercept to indicate that an average of ~93 amino acids are pre-unfolded during $\Delta t_1$ = 600 s. Consistently, the plot of pre-translocated distance vs. $\Delta t_1$, in *Figure 3E*, shows that the average of the pre-translocated distance across three replicates collected with $\Delta t_1$ = 600 s is ~97 amino acids. This result suggests that ~97 amino acids are pre-unfolded or pre-translocated in the presence of ATPγS before mixing with ATP.

The simple relationship between the number of steps, *n*, and the substrate length, *L*, is n = *L*/*m*. However, due to pre-translocated distance during $\Delta t_1$, some of the total length of the substrate has already been translocated. To account for this, we define the reduced length, *L'* = (*L* − *C*), where *C* represents the number of amino acids pre-translocated for a given $\Delta t_1$. Thus, the number of steps, *n*, is related to reduced length as n = (*L* − *C*)/*m* = *L'*/*m*, see *Figure 3—figure supplement 2* for schematic.

Thus, for all of the length-dependent time-courses collected with variable $\Delta t_1$ the number of steps, *n* in *Equation 5* was replaced with (*L* − *C*)/*m*, where *C* was determined independently for each replicate, see Materials and Methods. The time-courses were simultaneously fit with *m* and $k_U$ as global parameters and $k_{end}$ as a local parameter.

The rate constant describing dissociation, $k_{end}$, was local to each time-course because we observed local variability in this phase. This is not surprising as this phase represents dissociation and fluorescence changes due to refolding. Although the impact on the signal due to dissociation is likely the same for each substrate, the influence of refolding on the signal is probably not the same because of the differences in total length.

The solid lines in *Figure 3B* represent the best-fit lines and indicate excellent agreement between the model and the experimental time-courses. From the global fit of each of three replicates for three $\Delta t_1$ values we found the unfolding rate constant, $k_U$, the kinetic step-size, *m*, and the overall rate, $mk_U$ to all be independent of $\Delta t_1$, see *Figure 4B–D*. As a function of $\Delta t_1$ we determined an average kinetic step-size of *m* = (56.5 ± 0.7) aa step$^{-1}$ indicating that ~57 amino acids are unfolded between two rate-limiting steps with unfolding rate constant, $k_U$ = (0.017 ± 0.002) s$^{-1}$, see *Table 2*. The product of *m* and $k_U$ yields the overall rate of $mk_U$ = (0.89 ± 0.09) aa s$^{-1}$, which is in excellent agreement with (0.9 ± 0.1) aa s$^{-1}$ determined from the peak time analysis in *Figure 3D*.

**Table 2.** Parameters obtained from global fitting the time-courses obtained from experiments in *Figure 3* and *Figure 4—figure supplement 2* to Scheme 1 in *Figure 4A*.

| Parameters from model-dependent analysis | 500 µM [ATPγS] and 500 µM [ATP] | 150 µM [ATPγS] and 500 µM [ATP] |
|---|---|---|
| *m* (aa) | 56.5 ± 0.7 | 58 ± 4 |
| $k_U$ (s$^{-1}$) | 0.017 ± 0.002 | 0.055 ± 0.005 |
| $mk_U$ (aa s$^{-1}$) | 0.89 ± 0.09 | 3.1 ± 0.2 |

Error represent the standard deviation determined from three replicates.

## Impact of the ATP:ATPγS ratio on processive unfolding and translocation

To test the impact of different ATP:ATPγS mixing ratios, experiments were carried out as schematized in *Figure 3A* but with a lower concentration of ATPγS. As in *Figure 3A*, 8 µM ClpB was loaded into Syringe 1 but 600 µM ATPγS and 200 nM RepA-Titin$_X$ substrate were loaded into Syringe 2. Syringe 3, again, contains 1 mM ATP and 40 µM α-casein. Thus, after the two mixing events, the final concentration of ATP and ATPγS were 500 and 150 µM, respectively, yielding an approximately 3:1 ATP:ATPγS mixing ratio.

*Figure 4—figure supplement 2A* shows a series of representative time-courses for all three RepA-Titin$_X$ substrates at fixed $\Delta t_1$ = 600 s. The relative fluorescence enhancement value is approximately two-fold lower and the time-courses exhibit larger fluctuations in the signal compared to time-courses collected at 1:1 ATP:ATPγS, compare *Figure 4—figure supplement 2A* to *Figure 3B*.

The amplitude of a single turnover experiment is directly proportional to the amount of bound enzyme or the extent of binding, $\overline{X}$, given by *Equation 1* (*Lucius et al., 2003*), where $[ClpB_6 - RepA - Titin_X]$ represents hexameric ClpB bound to the RepA-Titin$_X$ substrate and $[RepA - Titin_X]_{Total}$ represents the total amount of RepA-Titin$_X$ present in the experiment. In this single turnover experiment, the peak height is expected to be proportional to the amount of ClpB that arrived at the fluorophore after mixing with ATP. However, the amount of ClpB that made it to the fluorophore is defined by the processivity, $P$, given by *Equation 2*, raised to the power of the number of steps taken, n, where $k_U$ is as defined in Scheme 1 (see *Figure 4A*) and $k_d$ is the rate constant for dissociation at each intermediate (*Lucius et al., 2003*). Thus, the peak height is proportional to the amount of ClpB initially bound, $\overline{X}$, times the processivity, $P^n$, times a fluorescence output factor or $\overline{FXP^n}$, see *Equations 1 and 2*

$$\overline{X} = \frac{[ClpB_6 - RepA - Titin_X]}{[RepA - Titin_X]_{Total}} \tag{1}$$

$$P = \frac{k_U}{k_U + k_d} \tag{2}$$

The observed reduction in peak height upon reducing the [ATPγS] is consistent with less ClpB bound, lower processivity, or both as the impact on fluorescence due to PIFE is expected to be the same for both [ATPγS]. Less ClpB bound is expected as we have shown that the fraction of hexameric ClpB present in solution is a function of both ATPγS concentration and ClpB concentration (*Lin and Lucius, 2015*; *Lin and Lucius, 2016*). Thus, less hexamer is present and able to bind the substrate at 150 µM ATPγS compared to 500 µM ATPγS. However, reduced processivity at lower ATPγS may also be contributing to the lower amplitude.

To determine the rate of ClpB catalyzed protein unfolding in the presence of approximately 3:1 ATP:ATPγS we repeated the experiment at different values of $\Delta t_1$, plotted substrate length vs. peak time, and performed the same peak time analysis as in *Figure 3*, see *Figure 4—figure supplement 2B–D*.

Interestingly, the rate of translocation in the presence of approximately 3:1 ATP:ATPγS is (4.3 ± 0.2) aa s⁻¹, see *Figure 4—figure supplement 2C* and *Table 1*. This is ~5 times faster than the ~0.9 aa s⁻¹ in the presence of a 1:1 ATP:ATPγS, see *Table 1*. This suggests that although, on the one hand, ATPγS activates ClpB, ATPγS is also competing with ATP and slowing unfolding and/or translocation.

*Figure 4—figure supplement 2D* shows the intercept values (pre-translocated distance) from the fits in *Figure 4—figure supplement 2B* plotted as a function of $\Delta t_1$. The intercept, pre-translocated distance, increases with increasing time, $\Delta t_1$, again consistent with ATPγS-driven pre-unfolding/translocation. The rate is ~0.05 aa s⁻¹, which represents the rate of translocation in the presence of 150 µM ATPγS. Consistently, this is slower than the ~0.09 aa s⁻¹ in the presence of 500 µM ATPγS although the values are within error. Interestingly, the intercept in *Figure 4—figure supplement 2D* is ~67 amino acids in the presence of 150 µM ATPγS, which is larger than the ~48 amino acids in the presence of 500 µM ATPγS. However, as can be seen in *Table 1*, this number has large uncertainty, presumably, due to the long extrapolation back to $\Delta t_1$ = 0.

The time-courses in *Figure 4—figure supplement 2A* were also subjected to global analysis using Scheme 1 in *Figure 4A*. The resultant kinetic parameters as a function of $\Delta t_1$ are shown in *Figure 4—figure supplement 3A–C*. From this analysis we determined a kinetic step-size of $m$ = (58 ± 4) aa step⁻¹ indicating that at these lower ATPγS concentrations we detect ~58 amino acids to be unfolded

between two rate-limiting steps with unfolding rate constant, $k_U = (0.055 \pm 0.005)$ s$^{-1}$. The overall rate of unfolding, $mk_U = (3.1 \pm 0.2)$ aa s$^{-1}$, which is in reasonable agreement with $(4.3 \pm 0.2)$ aa s$^{-1}$ determined from the peak time analysis, see *Tables 1 and 2*.

## Discussion

Substantial evidence exists indicating that ClpA processively unfolds and translocates a polypeptide through its axial channel, out the other side of its hexameric ring structures, and into the proteolytic barrel of ClpP (*Weber-Ban et al., 1999*; *Reid et al., 2001*). Thus, by homology, it has been hypothesized that ClpB and Hsp104 must employ the same mechanisms as ClpA. However, this has been difficult to test, in part, because of the lack of covalent modification coupled to ClpB/Hsp104 catalyzed processing of protein substrates. This means that when ClpB/Hsp104 processes a protein substrate, they do not hand it off to an associated protease. Consequently, the substrate both enters and leaves the reaction without covalent modification.

### Evidence of processive protein unfolding and translocation catalyzed by ClpB

In acknowledgment of the difficulty in detecting unfolding and translocation without covalent modification, Weibezahn et al. embarked on a clever protein engineering strategy to test for threading through the axial channel of ClpB (*Weibezahn et al., 2004*). In their strategy, they constructed a variant of ClpB that included the IGL loop from ClpA. The IGL loop is responsible for the interaction between ClpA and ClpP. The rationale being that if ClpB was 'forced' to interact with the protease, ClpP, and if proteolytic fragments were detected then this must indicate that ClpB threads substrate through the axial channel and into ClpP for proteolysis in the same way as ClpA.

Weibezahn et al. reported mixing the intrinsically disordered protein, α-casein, ClpB with the IGL loops added (ClpB(IGL)), ClpP, and ATP and observed the loss of the band on an SDS–PAGE (sodium dodecyl sulfate polyacrylamide gel electrophoresis) gel that represented α-casein (*Weibezahn et al., 2004*). However, using their construct, we showed faster loss of full length α-casein when this experiment was performed in the absence of ATP or any other energy source (*Li et al., 2015b*), a control that was missing in the Weibezahn et al. paper.

Cleavage in the absence of ATP calls into question what is being detected with the ClpB(IGL) construct since one expects enzyme catalyzed protein translocation and subsequent threading into ClpP to be an energy requiring process. Indeed, degradation is observed in the presence of ATP but it is observed to be faster in the absence of ATP. Thus, one is left asking what proportion of the observed degradation is energy dependent vs. energy independent. Equally important, the loss of the α-casein band on a gel can occur from a single cleavage event. Thus, loss of the band does not reveal if ClpB processively threaded the substrate into ClpP and multiple cleavage events occurred or if only one cleavage event occurred.

Wickner and co-workers have shown that both ClpB and Hsp104 can induce a loss of GFP fluorescence when GFP contains the first 70 amino acids of the RepA protein at the N-terminus as a binding site (*Doyle et al., 2007b*). With those results in mind, we attempted to develop the stopped-flow approach reported here using RepA-GFP. However, we observed loss of fluorescence upon simply mixing ClpB/Hsp104 with RepA-GFP and ATPγS (data not shown). Under the impression that ATPγS would only support assembly and binding but not ClpB catalyzed protein unfolding, we interpreted those observations to indicate that, in the presence of only ATPγS, ClpB bound to the junction between unfolded RepA and GFP and melted sufficient secondary structure to result in cooperative unfolding of GFP and subsequent loss of fluorescence. After all, protein unfolding is often cooperative, thus the motor may only need to destabilize some of the folded region of GFP to induce complete unfolding (*Nagy, 2004*). This led to questions about how we would determine the extent to which the structured region must be unfolded before GFP fluorescence is extinguished in a single turnover experiment. Thus, the RepA-GFP construct was abandoned.

# Development of transient state kinetics approach to examine the elementary steps in enzyme catalyzed protein unfolding and translocation

Our objective has been to develop a single turnover stopped-flow method that would report on the elementary kinetic steps in a single-round of enzyme catalyzed protein unfolding and translocation. We have been seeking to quantify the elementary rate constants, kinetic step-sizes, and processivity for the protein disaggregating machines, ClpB and Hsp104. To this end, we previously applied the single turnover stopped-flow approach that used unfolded polypeptides ranging in length between 30 and 50 amino acids as well as truncations of the unstructured protein α-casein of lengths up to 127 amino acids. Based on the observations of *Doyle et al., 2007b* we used a mix of ATP and ATPγS to activate the motor as we have done here (*Li et al., 2015b*; *Durie et al., 2019*; *Durie et al., 2018*).

Importantly, we define protein unfolding and translocation as two kinetically distinct processes. Thus, when using unstructured polypeptide chains, we interpret the results to represent polypeptide translocation and not enzyme catalyzed protein unfolding. Using the stopped-flow approach with unstructured polypeptides we observed that both ClpB and Hsp104 proceeded through two rate-limiting steps before rapid dissociation, independent of the substrate length provided (*Li et al., 2015b*; *Durie et al., 2019*). From those observations we proposed that ClpB and Hsp104 are non-processive translocases on unstructured polypeptides, $P \sim 0.6$, see *Equation 2*. However, we acknowledged, for both enzymes that we could not rule out the possibility that we were only detecting two slow steps either before or after fast translocation and the rate of translocation was outside the milli-second temporal resolution of the stopped-flow.

Avellaneda et al. reported fast and processive translocation of mechanically unfolded maltose-binding protein (MBP) using optical tweezer measurements and a 'hyper-active' variant of ClpB that does not require KJE or ATPγS to be activated (*Oguchi et al., 2012*; *Avellaneda et al., 2020*). In their approach, they attached the N- and C-terminus of MBP to polystyrene beads and optically trapped each bead. The protein was then mechanically unfolded by increasing the distance between the two optically trapped beads and then 1 µM ClpB(Y503D) was added. Upon adding ClpB(Y503D) and ATP they report rapid reduction in the distance between the two beads. From this measurement, they report ClpB translocates on the unfolded polypeptide chain at rates of ~250 and 450 aa s⁻¹. As we have expressed elsewhere, it is not clear if these rates represent the activity of a single ClpB hexamer as those experiments are single molecule with respect to the stretched MBP protein and not ClpB (*Lin et al., 2022*). However, if correct, the values should be interpreted as a rate of translocation on an unfolded polypeptide chain and not a rate for protein unfolding.

Here, we report a rate of ~3.1 and 0.9 aa s⁻¹ for enzyme catalyzed protein unfolding of tandem repeats of the Titin I27 domain in the presence of 500 µM ATP and either 150 or 500 µM ATPγS, respectively. These rates are two orders of magnitude slower than what was reported by *Avellaneda et al., 2020*, consistent with translocation and protein unfolding being kinetically distinct processes. Indeed, the measurements reported here are in the presence of ATPγS but the increase in rate from ~0.9 to 3.1 with a reduction in ATPγS by a factor of 3.3 is not suggesting that the rate will increase by two orders of magnitude if ATPγS was absent. Rather, we hypothesize that what we are detecting is rate-limited protein unfolding followed by undetectably fast translocation up to the next folded region and then the process repeats. However, whether or not the rates of translocation between unfolding events are as fast as reported by Avellaneda et al. requires further testing (*Lin et al., 2022*).

The rate-limiting protein unfolding of the Titin I27 domains followed by rapid translocation has not been reported for ClpAP and ClpXP. Olivares et al., using optical tweezer approaches with similar Titin I27 domains used here, reported rate-limiting translocation after fast protein unfolding for both ATP-dependent proteases, ClpAP and ClpXP (*Olivares et al., 2017*). They observed a dwell time preceding protein unfolding of ~0.9 and ~0.8 s for ClpXP and ClpAP, respectively. The inverse of this can be taken as the rate constant for protein unfolding and would yield a value of ~1.2 s⁻¹, which is in good agreement with our observed rate constant of ~0.9–3 s⁻¹ depending on the ATP:ATPγS mixing ratio. For ClpB, we propose that the slow unfolding is followed by rapid translocation on the unfolded chain where translocation by ClpB must be much faster than for ClpAP and ClpXP.

Importantly, using single turnover stopped-flow experiments we showed that ClpAP and ClpA exhibited different translocation mechanisms including overall rate, rate constants, and kinetics step-sizes on unfolded polypeptide chains. We interpreted those differences to indicate that ClpP

allosterically impacts the translocation mechanism employed by ClpA (*Miller et al., 2013*; *Rajendar and Lucius, 2010*). Thus, ClpA and ClpAP should be considered different enzymes. Consequently, it is not surprising that ClpA and ClpB would exhibit marked differences in their mechanisms of protein unfolding and translocation.

## Interpretation of the kinetic step-size

From the model independent peak time analysis, we determined an overall rate of protein unfolding. However, the rate of protein unfolding is a convolution of the number of amino acids unfolded per step and the rate constant defining that step. From global analysis, we can deconvolute the rate constant and step-size from the overall rate thereby extracting additional information about the elementary mechanism.

From global fitting of the stopped-flow time-courses to an *n*-step sequential mechanism we report a kinetic step-size of ~57 and~58 amino acids per step at 1:1 and ~3:1 ATP:ATPγS, respectively. Importantly, the kinetic step-size represents the average number of amino acids unfolded or translocated between two rate-limiting steps. In contrast to the kinetic step-size, we would define a mechanical protein unfolding step-size as the number of amino acids physically unfolded. Similarly, we would define a mechanical translocation step-size as the distance the motor translocates on the polypeptide chain per step.

In some cases, the kinetic step-size and mechanical step-size may be the same. For example, we reported a kinetic step-size of translocation for ClpAP to be ~5 amino acids per step (*Miller et al., 2013*). From optical tweezer experiments it was later reported that ClpAP exhibited a mechanical translocation step-size of 4–8 amino acids per step (*Olivares et al., 2014*), indicating that the same process was being monitored in both experimental approaches.

Here, we conclude that it is unlikely that ClpB physically traverses 56–58 amino acids in a single step. So, what is the meaning of this kinetic step-size? The magnitude must be governed by the structural properties of both the motor and the substrate being unfolded and translocated. From cryo-EM structures of ClpB bound to unfolded polypeptide chains, ~26 amino acids span the axial channel of the hexameric ring (*Rizo et al., 2019*). It seems unlikely that the motor would traverse 26 amino acids in a single step but the full length of the axial channel or 26 aa could be considered an upper limit on mechanical stepping. Nevertheless, the large kinetic step-size observed here, taken with a substantially slower rate on a folded compared to an unfolded polypeptide, suggests rate-limited protein unfolding of ~60 or more amino acids followed by rapid translocation that is outside of the millisecond temporal resolution of the stopped-flow. ClpB must apply force to the folded structure that results in the cooperative collapse of ~60 amino acids or, more likely, a full Titin I27 domain of 98 amino acids followed by rapid translocation up to the next folded domain, see *Figure 5*.

In our constructs, each Titin I27 domain is 98 amino acids so a step-size of ~60 amino acids is less than one full domain. Unfolding of less than one repeat is unexpected as AFM studies aimed at unfolding tandem repeats of Titin I27 have shown that each domain unfolds cooperatively, that is without a partially unfolded intermediate state (*Oberhauser et al., 2001*). Moreover, optical tweezer experiments showed that both ClpAP and ClpXP induced cooperative unfolding of each Titin I27 domain (*Olivares et al., 2017*). Consequently, the observed kinetic step-size reported here may not represent the mechanical unfolding step-size. With those observations in mind, if protein unfolding is fully rate limiting, we would expect an upper limit on the kinetic step-size to be the length of a Titin I27 domain or ~98 amino acids. However, the kinetic step-size is predicted to be smaller than the total amount of structure unfolded if translocation is partially rate limiting with unfolding, which may be the case here at these sub-saturating ATP concentrations since both translocation and unfolding are ATP-driven processes. Again, this is because the kinetic step-size is the average number of amino acids unfolded/translocated between two rate-limiting steps. Thus, if multiple translocation steps are partially rate limiting with a single protein unfolding event, then we predict a reduced kinetic step-size compared to the mechanical unfolding step-size. Testing this hypothesis with a complete [ATP] dependence of the kinetic parameters is underway.

A mechanical translocation step-size of 2 amino acids per step has been proposed from many cryo-EM structures of AAA+ motors bound to polypeptide substrates (*Gates and Martin, 2020*; *Glynn et al., 2020*). This is based on the distance between polypeptide contacts across the axial channel of the motor. However, to our knowledge, an experimental determination of a mechanical

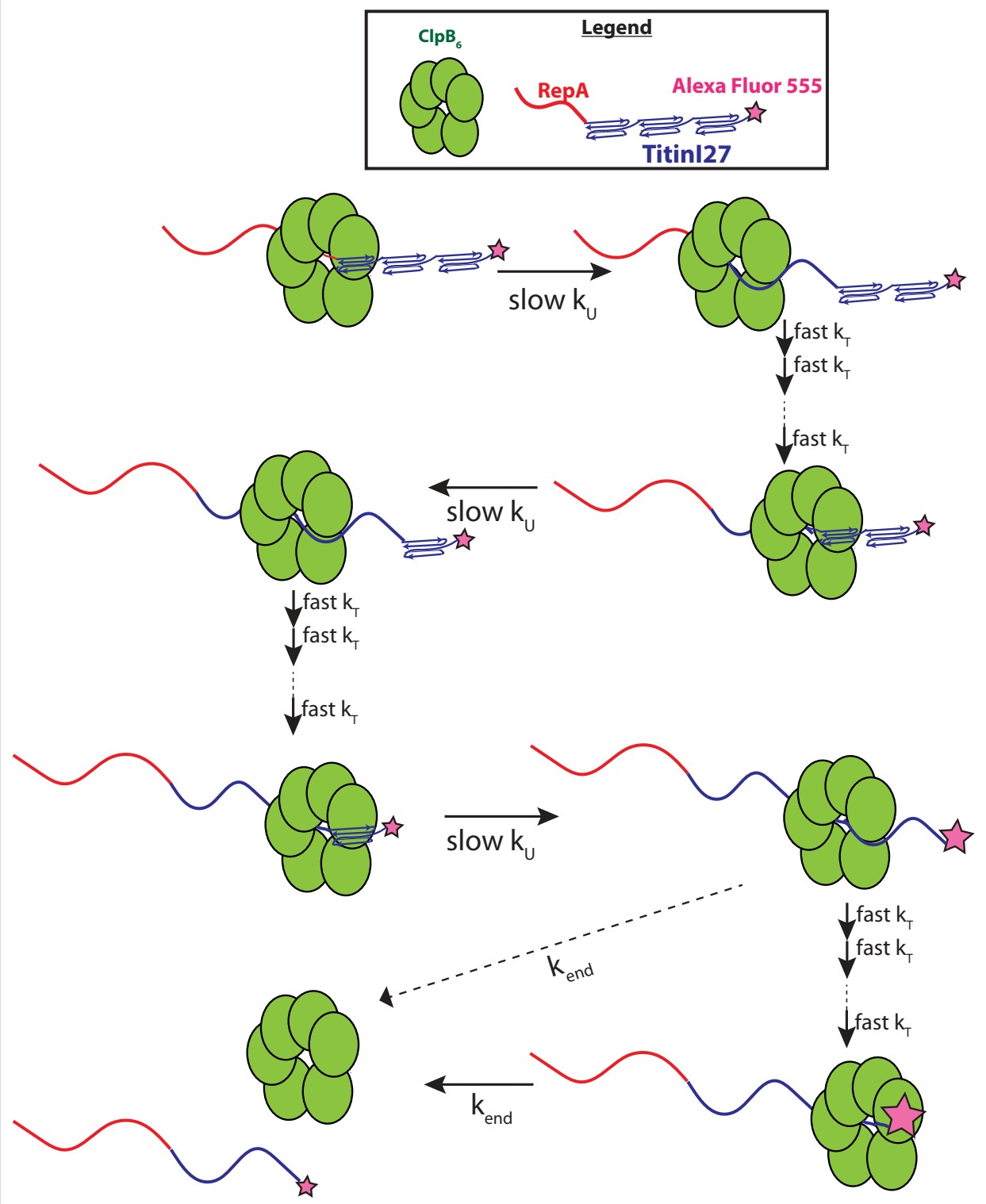

**Figure 5.** Proposed mechanism of ClpB catalyzed protein unfolding and translocation. In the presence of ATPγS, hexameric ClpB binds to the unfolded 70 amino acid long RepA sequence. Upon mixing with ATP, ClpB unfolds ~60 amino acids with rate constant, $k_U$, shown as complete collapse of the first Titin I27 domain. ClpB then proceeds through multiple fast translocation steps with rate constant $k_T$ before arrival at the next folded Titin I27 domain and the process repeats. After complete unfolding of the final Titin I27 domain, ClpB may dissociate before arrival at the C-terminus, or rapidly translocate to the C-terminus followed by slow dissociation with dissociation rate constant $k_{end}$. We expect protein-induced fluorescence enhancement (PIFE) to occur at the last unfolding step and/or at the last translocation step; the relative intensities of AF555 represented by the size of the pink star.

step-size as small as two amino acids per step has yet to be reported. Nevertheless, rate-limiting protein unfolding of a Titin I27 domain followed by fast translocation in small sub-steps remains a model to be tested.

## Quantification of unfolding vs. translocation processivity

The conclusion that ClpB is a non-processive translocase on unfolded polypeptide chains was based on the lack of any detectible length-dependence in the time-courses (*Li et al., 2015b*). The collected time-courses were most consistent with two-step dissociation. Here, we have shown that ClpB is able to unfold up to three tandem repeats of stably folded Titin I27 domains with a total substrate length of 362 amino acids. This indicates that, after unfolding the substrate, ClpB exhibits a translocation processivity of at least $N = 362$ amino acids, where $N$ is defined as the processivity in the number of amino acids translocated per binding event.

The processivity, $N$, in number of amino acids is related to the processivity as a probability, $P$, as defined in *Equation 3*, where m is the step-size (*Lucius et al., 2003*).

$$P = e^{-m/N} \tag{3}$$

Since the relationship between the probability processivity, $P$, and the number processivity, $N$, requires knowledge of the step-size, $m$, we can approximate the processivity, $P$, for translocation in the presence of fold to be in the range of $P = 0.994$ to $P = 0.946$ for a step-size between 2 and 20 amino acids, respectively, since we do not know the translocation step-size with certainty. In contrast, if we use a step-size of a full Titin I27 domain (98 amino acids), in *Equation 3* then $P = \sim 0.76$. Thus, these differences in processivity may suggest that when ClpB is challenged by stably folded regions the processivity for this process is reduced. Thus, going forward, it will be interesting to determine how processivity relates to stability of the folded regions. Equally, examining additional tandem repeats will also be of importance.

One explanation for the apparent disparity between this study and our previous work is that the previous lack of structure in the substrate may serve to signal ClpB to dissociate. If this were the case, then after ClpB unfolds the last Titin I27 domain and no more folded structure is down stream of the motor then one would expect ClpB to dissociate. However, given that the PIFE distance is 0–3 nm (*Hwang and Myong, 2014*), detecting PIFE is consistent with ClpB being within ~12 amino acids of the C-terminus before dissociation. This predicts that some number of translocation steps do occur on the unstructured polypeptide after complete unfolding of the final Titin I27 domain.

Another explanation is because ATPγS supports processive protein unfolding, it is possible that ATPγS-driven translocation obscured the observation of a length dependence in our previous studies. If ClpB was coupling ATPγS hydrolysis to processive rounds of translocation during the pre-incubation time in the stopped-flow then one would expect that, upon rapid mixing with ATP, the motor proteins would be randomly distributed on the polypeptide substrates. We know from studies on ssDNA translocases that random binding to the lattice to be translocated will still lead to the observation of a length dependence in a single turnover stopped-flow experiment (*Fischer and Lohman, 2004*; *Fischer et al., 2004*). Thus, if randomly bound motor on the peptides was the state of the system upon rapid mixing with ATP then we would predict the observation of a length dependence that we did not previously detect (*Li et al., 2015b*).

Alternatively, in the pre-incubation syringe, if ClpB used ATPγS to translocate up to the fluorophore and stalled at the end until rapid mixing with ATP that induced dissociation, then this situation could lead to the apparent non-processive translocation behavior previously reported. If this model is correct then the dissociation rate constant detected here, $k_{end}$, should be equivalent to the reported rate constants for dissociation from unstructured substrates. Here, we observe dissociation from the end to range from $k_{end} = 0.01–0.15$ s$^{-1}$ at 1:1 ATP:ATPγS and from our previous report we detected dissociation from the unstructured substrates to be between 0.03 and 0.15 s$^{-1}$. These numbers are in good agreement and may indicate that, in our previous study, we only detected dissociation from the end due to a stalled motor at either the N- or C-terminus. However, the experiments performed in the presence of only ATPγS, shown in *Figure 2B*, show a loss in signal consistent with the eventual dissociation of ClpB after arrival at the fluorophore. Thus, the time-courses reported here are consistent with slow dissociation from the end.

We propose that the best explanation is that ClpB rapidly translocates on unfolded polypeptide chains so fast that, on unfolded substrates, we only detect two slow dissociation steps. With the agreement between the rates of dissociation from the end of the substrates reported here and the values previously reported from unstructured polypeptides (*Li et al., 2015b*), we favor a model where rapid translocation is followed by slow dissociation from the end, see $k_{end}$ in *Figure 5*. This interpretation is most consistent with recent reports on unfolded polypeptides (*Avellaneda et al., 2020*; *Mazal et al., 2019*; *Mazal et al., 2021*).

More work is needed to fully understand the mechanisms of ClpB catalyzed protein unfolding and translocation. Several methods have been developed to examine the reactions catalyzed by ClpB in the absence of the co-chaperones, KJE. Here, we have used a mixture of ATP and ATPγS, which is one of the methods (*Doyle et al., 2007b*). However, applying our approach to the hyper-active variants is of interest, for example ClpB(Y503D). In addition, further development of the sequential mixing approach that will allow us to include the full complement of co-chaperones, KJE, is underway (*Durie et al., 2018*).

Here, we interpret our results to indicate that protein unfolding is rate-limiting and translocation is fast, see *Figure 5*. This predicts differences in the rate and kinetic step-size depending on the stability of the fold presented to the motor. With the establishment of the sequential mixing approach developed here we are well positioned to address this question. Moreover, we are now positioned to address questions on directionality and processivity that have been out of our reach for motors like ClpB and Hsp104. Furthermore, by interrogating the impact of protein stability on mechanical unfolding rates, rate constants, and step-sizes we may acquire insights into protein folding that complements common techniques such as chemical denaturation and heat denaturation. Moreover, how ATP hydrolysis is coupled to the reaction detected and the role of Domain 1 and 2 ATP binding and hydrolysis sites remain to be interrogated. Nevertheless, the work presented here has opened the door to answering these questions and many more about the fundamental reactions catalyzed by protein disaggregating machines and their role in protein homeostasis. With that in mind, so long as one can define a binding sequence, this approach can be broadly applied to the many AAA+ molecular motors that do not modify the substrates they process.

## Materials and methods
### Buffers and reagents

Buffers were prepared with reagent-grade chemicals using 18 MΩ deionized water from a Purelab Ultra Genetic system (Evoqua, Warrendale, PA). Buffer H200 contains 25 mM HEPES (2-[4-(2-hydroxyethyl) piperazin-1-yl]ethanesulfonic acid), pH = 7.5 at 25°C, 10 mM MgCl$_2$, 200 mM NaCl, 2 mM 2-mercaptoethanol, and 10% (vol/vol) glycerol. *E. coli* ClpB was purified as described (*Hwang et al., 2011*). All ClpB concentrations are reported in monomer units. ATP and ATPγS were purchased from Thermo Fisher Scientific (Waltham, MA) and CalBiochem (La Jolla, CA), respectively. Both ATP and ATPγS were dialyzed into H200 using 100–500 Da molecular weight cutoff dialysis tubing (Thermo Fisher Scientific, Waltham, MA). α-Casein was purchased from Sigma-Aldrich (Darmstadt, Germany), dissolved in 6 M guanidine hydrochloride, 20 mM HEPES, pH 7 at 25°C, and dialyzed into H200 using 10 kDa molecular weight cutoff dialysis tubing (Thermo Fisher Scientific, Waltham, MA).

### Purification of RepA-Titin$_X$

The His$_6$-RepA-Titin$_X$ proteins are composed of an N-terminal 6 His tag with a thrombin cleavage sequence for tag removal (MGSSHHHHHH SSGLVPRGSH). The 6 His tag is followed by the first 70 amino acids of the phage P1 RepA protein (MNQSFISDIL YADIESKAKE LTVNSNNTVQ PVALM-RLGVF VPKPSKSKGE SKEIDATKAF SQLEIAKAEG) (*Kenniston et al., 2003*). For RepA-Titin$_1$, the RepA sequence is directly connected to the Titin I27 domain (PDB ID: 2RQ8) (*Durie et al., 2018*) with all native cysteine residues changed to alanine and the final sequence is given by MLIEVEKPLY GVEVFVGETA HFEIELSEPD VHGQWKLKGQ PLAASPDAEI IEDGKKHILI LHNAQLGMTG EVSFQAANT KSAANLKVKE L. Additional Titin I27 domains are connected with an eight residue linker, RSKLGTRM. Each construct contains a single cysteine at the C-terminus for fluorescent modification. The genes encoding for RepA(1-70)Titin$_X$ were constructed and cloned into the pET28a vector by Genscript

(Piscataway, NJ). The total substrate lengths are 168, 265, and 362 amino acids for RepA-Titin$_1$, RepA-Titin$_2$, and RepA-Titin$_3$, respectively.

The pET28a plasmids were transformed into ΔClpP$_{WT}$-BL21 cells using electroporation (**Seidman et al., 2001**). ClpP$_{WT}$ knockout BL21(DE3) cells, ΔClpP$_{WT}$-BL21, were constructed using recombineering (**Sharan et al., 2009**). ClpP$_{WT}$ gene in BL21(DE3) genome was substituted with an Ampicillin cassette similar to what has been done earlier with ΔClpA$_{WT}$-BL21(DE3) cells (**Duran and Lucius, 2018**). The ΔClpP$_{WT}$-BL21 cells were grown in LB Miller growth media (Thermo Fisher Scientific, Waltham, MA) containing 100 µg/ml Ampicillin.

ΔClpP$_{WT}$-BL21 cells were grown to mid-log growth and induced with 1 mM IPTG to express His$_6$-RepA-Titin$_X$. The cell paste was suspended in cell lysis buffer [40 mM Tris pH 7.5 at 4°C, 20 mM Imidazole, 500 mM NaCl, 2 mM 2-mercaptoethanol, 10% (wt/vol) sucrose, 20% (vol/vol) glycerol]. Cells were lysed using a French Press at 4°C. DNase I (Thermo Fisher Scientific, Waltham, MA) and RNase A (Thermo Fisher Scientific, Waltham, MA) were added to the lysate and incubated at 4°C for 15 min. Cell debris was pelleted by centrifugation for 120 min at 28,000 × $g$ and 4°C in a ThermoScientific Fiberlite F14-6x250 rotor. His$_6$-RepA-Titin$_X$ was in the soluble fraction of the lysate.

Isolation of His$_6$-RepA-Titin$_X$ started with batch purification. The soluble fraction was added to HisPur Ni-NTA resin (Thermo Fisher Scientific, Waltham, MA) and incubated with rocking for 75 min at 4°C. The resin was subsequently washed with 5–10 column volumes of binding buffer (40 mM Tris, pH 7.5 at 4°C, 20 mM Imidazole, 500 mM NaCl, 2 mM 2-mercaptoethanol, 10% (vol/vol) glycerol). The resin was then incubated with 1 column volume of elution buffer (40 mM Tris, pH 7.5 at 4°C, 500 mM Imidazole, 500 mM NaCl, 2 mM 2-mercaptoethanol, 10% (vol/vol) glycerol) for 1 hr at 4°C on a rocker. The slurry, comprising the resin and the elution buffer was altogether poured into a gravity column and incubated for 10 min to form a resin bed. His$_6$-RepA-Titin$_X$ bound to the resin was expected to elute with the elution buffer that is under high Imidazole conditions. Fractions of 1 mL were collected from the gravity column. Further the resin bed was washed with 5–10 column volumes of elution buffer to elute any remaining His$_6$-RepA-Titin$_X$ from the resin. Fractions containing His$_6$-RepA-Titin$_X$ were identified using a NuPAGE gel (Thermo Fisher Scientific, Waltham, MA) with Coomassie staining. The fractions of interest were dialyzed into Thrombin digestion buffer (20 mM Tris, pH 7.5 at 4°C, 150 mM NaCl, 2.5 mM CaCl$_2$) using a 10 kDa cutoff dialysis tubing (Thermo Fisher Scientific, Waltham, MA).

His$_6$-RepA-Titin$_X$ was subjected to Thrombin protease (Sigma-Aldrich, Burlington, MA) following the manufacturer's protocol. The protein sample was passed through His Trap FF crude column (GE Healthcare, Piscataway, NJ). We expected the undigested protein to remain bound to the column. After washing the column with the digestion buffer, RepA-Titin$_X$ was washed out with the binding buffer. All the fractions of interest were pooled together and dialyzed into a low salt buffer (20 mM Tris, pH 7.5 at 4°C, 20 mM NaCl, 2 mM 2-mercaptoethanol). Furthermore, the protein sample was passed through a Hi-Trap Q-Sepharose FF crude (GE Healthcare, Piscataway, NJ) column. The column was washed with a linear gradient of low salt and high salt buffer (20 mM Tris, pH 7.5 at 4°C, 1 M NaCl, 2 mM 2-mercaptoethanol) starting with 100% low salt buffer (20 mM NaCl) and moving gradually to 100% high salt buffer (1 M NaCl). RepA-Titin$_X$ eluted at low salt conditions. The fractions of interest were dialyzed into labeling buffer (20 mM HEPES, pH = 7.5 at 4°C, 50 µM Tris(2-carboxyethyl)phosphine (TCEP)).

Labeling of C-terminal Cysteine on RepA-Titin$_X$ was carried out using a C$_2$-maleimide reaction with Alexa Fluor 555 fluorophore (Invitrogen, Waltham, MA) following the manufacturer's protocol except 3 (fluorophore): 1 (protein) molar excess. After labeling, RepA-Titin$_X$-Alexa Fluor 555 along with free Alexa Fluor 555 was dialyzed into the H200 buffer. Free Alexa Fluor 555 was then separated from RepA-Titin$_X$-Alexa Fluor 555 using a HiPrep 26/10 desalting column (GE Healthcare, Piscataway, NJ). The labeled protein sample was dialyzed into storage buffer (40 mM Tris, 500 mM NaCl, 2 mM 2-mercaptoethanol, 10% (vol/vol) glycerol, 2 mM Ethylenediaminetetraacetic acid(EDTA)) and stored at −80°C. The labeling efficiency of RepA-Titin$_X$-Alexa Fluor 555 in H200 buffer was observed to be 65–100%. The reported concentrations of RepA-Titin$_X$ substrates are determined spectrophotometrically by measuring the absorbance of Alexa Fluor 555 at 555 nm and using an extinction coefficient of 158 000 M$^{-1}$ cm$^{-1}$.

## Structures of RepA-Titin$_X$

The raw structures of RepA-Titin$_X$ with X = 1, 2, or 3 were obtained from simulations using Alpha-Fold (*Jumper et al., 2021*). Starting from the N-terminus, the first 70 amino acids represent the N-terminus of the Phage P1 RepA protein. AlphaFold predicted low confidence on α helices present in the RepA 1–70 sequence. This is consistent with previous reports that this sequence is not likely structured (*Hoskins et al., 2000*; *Sharma et al., 2004*; *Kim et al., 2002*). Thus, we used 'Sculpting' function in *Schrodinger, 2015a*; *Schrodinger, 2015b* to unfold the α helices in the RepA sequence in the AlphaFold predicted structures to yield the structures shown in *Figure 1A*. The TitinI27 regions are predicted to have folded β sandwich structure consistent with the published structure (PDB ID: 2RQ8) (*Yagawa et al., 2010*). The connectors between tandem repeats of TitinI27 are predicted to be unstructured as shown in *Figure 1*. Cysteine is shown in space-filling at the C terminus of each substrate. Alexa Fluor 555 is attached to the C-terminal cysteine, not shown.

## Standard mixing stopped-flow experiments

ClpB and RepA-Titin$_X$ were dialyzed into H200 buffer using 50 and 10 kDa molecular weight cutoff dialysis tubing, respectively. The experiments were performed on an SX20 Applied Photophysics stopped-flow fluorometer (Leatherhead, UK) under standard mixing set-up at 25°C as shown in *Figure 1C*. 1.5 μM ClpB was incubated for 5 min with 300 μM ATPγS to form the active hexameric complex and subsequently incubated for 10 min with 100 nM RepA-Titin$_X$ substrate to form the pre-bound complex. The pre-bound complex was loaded into Syringe 1 and 400 μM ATP and 20 μM α-casein into Syringe 2 of the apparatus. All the components are diluted twofold upon mixing. Before collection of the first shot, the observation channel was thoroughly washed with the solutions of each syringe to equilibrate the instrument. For comparing the relative increase in fluorescence signal across RepA-Titin$_X$ substrates, processing of raw time-courses were done using *Equation 4*.

$$\text{Relative fluorescence enhancement} = \frac{\mid \left( F_o^{av} - F_t \right) \mid}{F_o^{av}} \tag{4}$$

where $F_o^{av}$ is the average of the first few constant raw fluorescence data points and $F_t$ is raw fluorescence signal at a given time.

## Sequential mixing stopped-flow experiments

8 μM ClpB was rapidly mixed with 2000 μM ATPγS and 200 nM RepA-Titin$_X$ substrate. After mixing, the reagents incubate in the ageing loop for a user-defined period of time, $\Delta t_1$. 1000 μM ATP and 40 μM α-casein in Syringe 3 were rapidly mixed with the pre-bound complex of ClpB with RepA-Titin$_X$ formed during sequential mixing in ageing loop, see *Figure 3A*. All the components in Syringes 1 and 2 are diluted fourfold when present in the observation chamber while components in Syringe 3 are diluted twofold. In the observation channel, Alexa Fluor 555 is excited at 555 nm and the emission signal is collected using a 570-nm long pass filter.

## Determination of reduced length

For each set of time-courses collected using RepA-Titin$_X$, we plot total length vs. peak time. The intercept of this plot is $C$, which is the sum of pre-translocated distance with ATPγS and excluded length. Excluded length is defined as the sum of dangle distance and occluded length, see *Figure 3—figure supplement 2*. So, we subtract $C$ from total length, $L$ to get reduced length of substrate, $L'$. $L'$ is the available length for ClpB to translocate and unfold. To note, $C$ is determined individually for each replicate at a particular $\Delta t_1$. The average values of $C$ at different $\Delta t_1$'s are shown in *Figure 3E* and *Figure 4—figure supplement 2D*.

## Model-dependent analysis of the experimental time-courses

The time-courses were fit using the custom built MATLAB (Mathworks, Natick, MA) toolbox, MENOTR (*Ingram et al., 2021*). *Equation 5* described the dataset for each RepA-Titin$_X$ and the resultant kinetic parameters are described in *Table 2*. Out of all parameters, $k_U$ and $m$ were fit globally across each RepA-Titin$_X$.

$$F(t) = \mathcal{L}^{-1} \left\{ \frac{F_1(k_u)^n}{(k_{end} + S)(k_u + S)^n} + \frac{F_2 k_{end}(k_u)^n}{S(k_{end} + S)(k_u + S)^n} \right\} \qquad (5)$$

where $F_1$ and $F_2$ are two fluorescence amplitudes for the last intermediate, $I_{(L-nm)}$ and unfolded RepA-Titin$_x$, respectively, $s$ is the Laplace variable and other kinetic parameters are defined as per Scheme 1, see **Figure 4A**.

## Acknowledgements

We thank Elizabeth Duran, Clarissa Durie, and members of the Lucius lab for their critical discussions of the results and the manuscript. This work was supported by the National Science Foundation (grant NSF MCB-1412624 to ALL). Computational work was performed using the UAB High Performance Computing (HPC) Cheaha, which is supported in part by the National Science Foundation under Grant No. OAC-1541310, the University of Alabama at Birmingham, and the Alabama Innovation Fund.

## Additional information

### Competing interests

Jaskamaljot Kaur Banwait, Aaron L Lucius: consultant for Nitrase Therapeutics. The other author declares that no competing interests exist.

### Funding

| Funder | Grant reference number | Author |
| --- | --- | --- |
| National Science Foundation | MCB-1412624 | Aaron L Lucius |
| National Science Foundation | OAC-1541310 | Aaron L Lucius |

The funders had no role in study design, data collection, and interpretation, or the decision to submit the work for publication.

### Author contributions

Jaskamaljot Kaur Banwait, Conceptualization, Resources, Data curation, Formal analysis, Investigation, Visualization, Methodology, Writing – original draft, Project administration, Writing – review and editing; Liana Islam, Resources; Aaron L Lucius, Conceptualization, Resources, Formal analysis, Supervision, Funding acquisition, Validation, Investigation, Methodology, Writing – original draft, Project administration, Writing – review and editing

### Author ORCIDs

Jaskamaljot Kaur Banwait ⓘD http://orcid.org/0009-0005-7494-1588
Liana Islam ⓘD https://orcid.org/0009-0005-5284-9545
Aaron L Lucius ⓘD https://orcid.org/0000-0001-8636-5411

Reviewer #1 (Public review): https://doi.org/10.7554/eLife.99052.3.sa1
Reviewer #2 (Public review): https://doi.org/10.7554/eLife.99052.3.sa2
Reviewer #3 (Public review): https://doi.org/10.7554/eLife.99052.3.sa3
Author response https://doi.org/10.7554/eLife.99052.3.sa4

## Additional files

### Supplementary files
• MDAR checklist

## Data availability

Figure 1—source data 1, Figure 2—source data 1, Figure 3—source data 1 and Figure 4—source data 1 contain the numerical data used to generate Figures 1–4, respetivley. The same naming convention is used for the supplemental data.

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
