## [Editor Report · eLife assessment]

This **valuable** study presents the development of a single turnover stopped-flow fluorescence experiment to study the kinetics of substrate unfolding and translocation by the bacterial ClpB disaggregase. Using non-physiological nucleotides to bypass the physiological regulation mechanism of ClpB, the authors **convincingly** show that the ClpB disaggregase is a processive motor with a slow unfolding step preceding rapid translocation. The results of this analysis are of value for future mechanistic studies on energy-dependent unfolding, degradation, and disaggregation molecular machines.

---

## [Referee Report · Reviewer #1 (Public review)]

In this study, the Authors used a stopped-flow method to investigate the kinetics of substrate translocation through the channel in hexameric ClpB, an ATP-dependent bacterial protein disaggregase. They engineered a series of polypeptides with the N-terminal RepA ClpB-targeting sequence followed by a variable number of folded titin domains. The Authors detected translocation of the substrate polypeptides by observing the enhancement of fluorescence from a probe located at the substrate's C-terminus. The total time of the substrates' translocation correlated with their lengths, which allowed the Authors to determine the number of residues translocated by ClpB per unit time.

Strengths:

This study confirms a previously proposed model of processive translocation of polypeptides through the channel in ClpB. The novelty of this work is in a clever design of a series of kinetic experiments with an engineered substrate that includes stably folded domains. This approach produced a quantitative description of the reaction rates and kinetic step sizes. Another valuable aspect is that the method can be used for other translocases from the AAA+ family to characterize their mechanism of substrate processing.

Weaknesses:

The main limitation of the study is in using a single non-physiological substrate of ClpB, which does not replicate physical properties of the aggregated cellular proteins and includes a non-physiological ClpB-targeting sequence. Another limitation is in the use of ATPgammaS to stimulate the substrate processing. It is not clear how relevant the results are to the ClpB function in living cells with ATP as the source of energy, a multitude of various aggregated substrates without targeting sequences that need ClpB's assistance, and in the presence of the co-chaperones.

Evidence that ATPgammaS without ATP can provide sufficient energy for substrate translocation and unfolding is missing in the paper because the rate of phosphate release from ATPgammaS has not been determined. Thus, it is not clear if the observed translocation is linked to an actual chemical energy input or is a result of a diffusion-driven ratchet mediated by a substrate-trapping ClpB conformation obtained in the presence of ATPgammaS.

---

## [Referee Report · Reviewer #2 (Public review)]

Summary:

The current work by Banwait et al. reports a fluorescence-based single turnover method based on protein-induced fluorescence enhancement (PIFE) to show that ClpB is a processive motor. The paper is a crucial finding as there has been ambiguity on whether ClpB is a processive or non-processive motor. Optical tweezers-based single-molecule studies have shown that ClpB is a processive motor, whereas previous studies from the same group hypothesized it to be a non-processive motor. As co-chaperones are needed for the motor activity of the ClpB, to isolate the activity of ClpB, they have used a 1:1 ratio ATP and ATPgS, where the enzyme is active even in the absence of its co-chaperones, as previously observed. A sequential mixing stop-flow protocol was developed, and the unfolding and translocation of RepA-TitinX, X = 1,2,3 repeats was monitored by measuring the fluorescence intensity with time of Alexa F555 that was labelled at the C-terminal Cysteine. The observations were a lag time, followed by a gradual increase in fluorescence due to PIFE, and then a decrease in fluorescence plausibly due to the dissociation from the substrate allowing it to refold. The authors observed that the peak time depends on the substrate length, indicating the processive nature of ClpB. In addition, the lag and peak times depend on the pre-incubation time with ATPgS, indicating that the enzyme translocates on the substrates even with just ATPgS without the addition of ATP, which is plausible due to the slow hydrolysis of ATPgS. From the plot of substrate length vs peak time, the authors calculated the rate of unfolding and translocation to be ~0.1 aas-1 in the presence of ~1 mM ATPgS and increases to 1 aas-1 in the presence of 1:1 ATP and ATPgS. The authors have further performed experiments at 3:1 ATP and ATPgS concentrations and observed ~5 times increase in the translocation rates as expected due to faster hydrolysis of ATP by ClpB and reconfirming that processivity is majorly ATP driven. Further, the authors model their results to multiple sequential unfolding steps, determining the rate of unfolding and the number of amino acids unfolded during each step. Overall, the study uses a novel method to reconfirm the processive nature of ClpB.

Strengths:

(1) Previous studies on understanding the processivity of ClpB have primarily focused on unfolded or disordered proteins; this study paves new insights into our understanding of the processing of folded proteins by ClpB. They have cleverly used RepA as a recognition sequence to understand the unfolding of titin-I27 folded domains.

(2) The method developed can be applied to many disaggregating enzymes and has broader significance.

(3) The data from various experiments are consistent with each other, indicating the reproducibility of the data. For example, the rate of translocation in presence of ATPgS, ~0.1 aas-1 from the single mixing experiment and double mixing experiment are very similar.

(4) The study convincingly shows that ClpB is a processive motor, which has long been debated, describing its activity in the presence of only ATPgS and a mixture of ATP and ATPgS.

(5) The discussion part has been written in a way that describes many previous experiments from various groups supporting the processive nature of the enzyme and supports their current study.

Weaknesses:

(1) The authors model that the enzyme unfolds the protein sequentially around 60 aa each time through multiple steps and translocates rapidly. This contradicts our knowledge of protein unfolding, which is generally cooperative, particularly for titinI27, which is reported to unfold cooperatively or utmost through one intermediate during enzymatic unfolding by ClpX and ClpA.

(2) It is also important to note that the unfolding of titinI27 from the N-terminus (as done in this study) has been reported to be very fast and cannot be the rate-limiting step as reported earlier(Olivares et al, PNAS, 2017). This contradicts the current model where unfolding is the rate-limiting step, and the translocation is assumed to be many orders faster than unfolding.

(3) The model assumes the same time constant for all the unfolding steps irrespective of the secondary structural interactions.

(4) Unlike other single-molecule optical tweezer-based assays, the study cannot distinguish the unfolding and translocation events and assumes that unfolding is the rate-limiting step.

---

## [Referee Report · Reviewer #3 (Public review)]

Summary:

The authors have devised an elegant stopped flow fluorescence approach to probe the mechanism of action of the Hsp100 protein unfoldase ClpB on an unfolded substrate (RepA) coupled to 1-3 repeats of a folded titin domain. They provide useful new insight into the kinetics of ClpB action. The results support their conclusions for the model setup used.

Strengths:

The stopped flow fluorescence method with a variable delay after mixing the reactants is informative, as is the use of variable numbers of folded domains to probe the unfolding steps.

Weaknesses:

The setup does not reflect the physiological setting for ClpB action. A mixture of ATP and ATPgammaS is used to activate ClpB without the need for its co-chaperones, Hsp70. Hsp40 and an Hsp70 nucleotide exchange factor. This nucleotide strategy was discovered by Doyle et al (2007) but the mechanism of action is not fully understood. Other authors have used different approaches. As mentioned by the authors, Weibezahn et al used a construct coupled to the ClpA protease to demonstrate translocation. Avellaneda et al used a mutant (Y503D) in the coiled coil regulatory domain to bypass the Hsp70 system. These differences complicate comparisons of rates and step sizes with previous work. It is unclear which results, if any, reflect the in vivo action of ClpB on disassembly of aggregates.

---

## [Author Response]

The following is the authors’ response to the original reviews.

**Public Reviews:**

**Reviewer #1 (Public Review):**
In this study, the authors used a stopped-flow method to investigate the kinetics of substrate translocation through the channel in hexameric ClpB, an ATP-dependent bacterial protein disaggregase. They engineered a series of polypeptides with the N-terminal RepA ClpB-targeting sequence followed by a variable number of folded titin domains. The authors detected translocation of the substrate polypeptides by observing the enhancement of fluorescence from a probe located at the substrate's C-terminus. The total time of the substrates' translocation correlated with their lengths, which allowed the authors to determine the number of residues translocated by ClpB per unit time.Strengths:This study confirms a previously proposed model of processive translocation of polypeptides through the channel in ClpB. The novelty of this work is in the clever design of a series of kinetic experiments with an engineered substrate that includes stably folded domains. This approach produced a quantitative description of the reaction rates and kinetic step sizes. Another valuable aspect is that the method can be used for other translocases from the AAA+ family to characterize their mechanism of substrate processing.Weaknesses:The main limitation of the study is in using a single non-physiological substrate of ClpB, which does not replicate the physical properties of the aggregated cellular proteins and includes a non-physiological ClpB-targeting sequence. Another limitation is in the use of ATPgammaS to stimulate the substrate processing. It is not clear how relevant the results are to the ClpB function in living cells with ATP as the source of energy, a multitude of various aggregated substrates without targeting sequences that need ClpB's assistance, and in the presence of the co-chaperones.

Indeed, we agree that our RepA-Titinx substrates are not aggregates but are model, soluble, substrates used to reveal information about enzyme catalyzed protein unfolding and translocation. Our substrates are similar to RepA-GFP and GFP-SsrA used by multiple labs including Wickner, Horwich, Sauer, Baker, Shorter, Bukua, to name only a few. The fact that “this is what everyone does” does not make the substrates physiological or the most ideal. However, this is the technology we currently have until we and others develop something better. In the meantime, we contend that the results presented here do advance our knowledge on enzyme catalyzed protein unfolding

Part of what this manuscript seeks to accomplish is presenting the development of a single-turnover experiment that reports on processive protein unfolding by AAA+ molecular motors, in this case, ClpB. Importantly, we are treating translocation on an unfolded polypeptide chain and protein unfolding of stably folded proteins as two distinct reactions catalyzed by ClpB. If these functions are used to disrupt protein aggregates, in vivo, then this remains to be seen.

We contend that processive ClpB catalyzed protein unfolding has not been rigorously demonstrated prior to our results presented here. Avellaneda et al mechanically unfolded their substrate before loading ClpB (Avellaneda, Franke, Sunderlikova et al. 2020). Thus, their experiment represents valuable observations reflecting polypeptide translocation on a pre-unfolded protein. Our previous work using single-turnover stopped-flow experiments employed unstructured synthetic polypeptides and therefore reflects polypeptide translocation and not protein unfolding (Li, Weaver, Lin et al. 2015). Weibezahn et al used unstructured substrates in their study with ClpB (BAP/ClpP), and thus their results represent translocation of a pre-unfolded polypeptide and not enzyme catalyzed protein unfolding (Weibezahn, Tessarz, Schlieker et al. 2004).

Many studies have reported the use of GFP with tags or RepA-GFP and used the loss of GFP fluorescence to conclude protein unfolding. However, such results do not reveal if ClpB processively and fully translocates the substrate through its axial channel. One cannot rule out, even when trapping with “GroEL trap”, the possibility that ClpB only needs to disrupt some of the fold in GFP before cooperative unfolding occurs leading to loss of fluorescence. Once the cooperative collapse of the structure occurs and fluorescence is lost it has not been shown that ClpB will continue to translocate on the newly unfolded chain or dissociate. In fact, the Bukau group showed that folded YFP remained intact after luciferase was unfolded (Haslberger, Zdanowicz, Brand et al. 2008). Our approach, reported here, yields signal upon arrival of the motor at the c-terminus or within the PIFE distance thus we can be certain that the motor does arrive at the c-terminus after unfolding up to three tandem repeats of the Titin I27 domain.

ATPgS is a non-physiological nucleotide analog. However, ClpB has been shown to exhibit curious behavior in its presence that we and others, as the reviewer acknowledges, do not fully understand (Doyle, Shorter, Zolkiewski et al. 2007). Some of the experiments reported here are seeking to better understand that fact. Here we have shown that ATPgS alone will support processive protein unfolding. With this assay in hand, we are now seeking to go forward and address many of the points raised by this reviewer.

The authors do not attempt to correlate the kinetic step sizes detected during substrate translocation and unfolding with the substrate's structure, which should be possible, given how extensively the stability and unfolding of the titin I27 domain were studied before. Also, since the substrate contains up to three I27 domains separated with unstructured linkers, it is not clear why all the translocation steps are assumed to occur with the same rate constant.

We assume that all protein unfolding steps occur with the same rate constant, ku. We conclude that we are not detecting the translocation rate constant, kt, as our results support a model where kt is much faster than ku. We do think it makes sense that the same slow step occurs between each cycle of protein unfolding.

We have added a discussion relating our observations to mechanical unfolding of tandem repeats of Titin I27 from AFM experiments (Oberhauser, Hansma, Carrion-Vazquez and Fernandez 2001). Most interestingly, they report unfolding of Titin I27 in 22 nm steps. Using 0.34 nm per amino acids this yields ~65 amino acids per unfolding step, which is comparable to our kinetic step-size of 57 – 58 amino acids per step.

Some conclusions presented in the manuscript are speculative:The notion that the emission from Alexa Fluor 555 is enhanced when ClpB approaches the substrate's C-terminus needs to be supported experimentally. Also, evidence that ATPgammaS without ATP can provide sufficient energy for substrate translocation and unfolding is missing in the paper.

In our previous work we have used fluorescently labeled 50 amino acid peptides as substrates to examine ClpB binding (Li, Lin and Lucius 2015, Li, Weaver, Lin et al. 2015). In that work we have used fluorescein, which exhibits quenching upon ClpB binding. We have added a control experiment where we have attached alexa fluor 555 to the 50 amino acid substrate so we can be assured the ClpB binds close to the fluorophore. As seen in supplemental Fig. 1 A upon titration with ClpB, in the presence of ATPγS, we observe an increase in fluorescence from AF555, consistent with PIFE. Supplemental Fig. 1 B shows the relative fluorescence enhancement at the peak max increases up to ~ 0.2 or a 20 % increase in fluorescence, due to PIFE, upon ClpB binding.

Further, peak time is our hypothesized measure of ClpB’s arrival at the dye. Our results indicate that the peak time linearly increases as a function of an increase in the number of folded TitinI27 repeats in the substrates which also supports the PIFE hypothesis. Finally, others have shown that AF555 exhibits PIFE and we have added those references.

The evidence that ATPγS alone can support translocation is shown in Fig. 2 and supplemental Figure 1. Fig. 2 and supplemental Figure 1 are two different mixing strategies where we use only ATPgS and no ATP at all. In both cases the time courses are consistent with processive protein unfolding by ClpB with only ATPγS.

**Reviewer #2 (Public Review):**
Summary:The current work by Banwait et al. reports a fluorescence-based single turnover method based on protein-induced fluorescence enhancement (PIFE) to show that ClpB is a processive motor. The paper is a crucial finding as there has been ambiguity on whether ClpB is a processive or non-processive motor. Optical tweezers-based single-molecule studies have shown that ClpB is a processive motor, whereas previous studies from the same group hypothesized it to be a non-processive motor. As co-chaperones are needed for the motor activity of the ClpB, to isolate the activity of ClpB, they have used a 1:1 ratio ATP and ATPgS, where the enzyme is active even in the absence of its co-chaperones, as previously observed. A sequential mixing stop-flow protocol was developed, and the unfolding and translocation of RepA-TitinX, X = 1,2,3 repeats was monitored by measuring the fluorescence intensity with the time of Alexa F555 which was labelled at the C-terminal Cysteine. The observations were a lag time, followed by a gradual increase in fluorescence due to PIFE, and then a decrease in fluorescence plausibly due to the dissociation from the substrate allowing it to refold. The authors observed that the peak time depends on the substrate length, indicating the processive nature of ClpB. In addition, the lag and peak times depend on the pre-incubation time with ATPgS, indicating that the enzyme translocates on the substrates even with just ATPgS without the addition of ATP, which is plausible due to the slow hydrolysis of ATPgS. From the plot of substrate length vs peak time, the authors calculated the rate of unfolding and translocation to be ~0.1 aas-1 in the presence of ~1 mM ATPgS and increases to 1 aas-1 in the presence of 1:1 ATP and ATPgS. The authors have further performed experiments at 3:1 ATP and ATPgS concentrations and observed ~5 times increase in the translocation rates as expected due to faster hydrolysis of ATP by ClpB and reconfirming that processivity is majorly ATP driven. Further, the authors model their results to multiple sequential unfolding steps, determining the rate of unfolding and the number of amino acids unfolded during each step. Overall, the study uses a novel method to reconfirm the processive nature of ClpB.Strengths:(1) Previous studies on understanding the processivity of ClpB have primarily focused on unfolded or disordered proteins; this study paves new insights into our understanding of the processing of folded proteins by ClpB. They have cleverly used RepA as a recognition sequence to understand the unfolding of titin-I27 folded domains.(2) The method developed can be applied to many disaggregating enzymes and has broader significance.(3) The data from various experiments are consistent with each other, indicating the reproducibility of the data. For example, the rate of translocation in the presence of ATPgS, ~0.1 aas-1 from the single mixing experiment and double mixing experiment are very similar.(4) The study convincingly shows that ClpB is a processive motor, which has long been debated, describing its activity in the presence of only ATPgS and a mixture of ATP and ATPgS.(5) The discussion part has been written in a way that describes many previous experiments from various groups supporting the processive nature of the enzyme and supports their current study.Weaknesses:(1) The authors model that the enzyme unfolds the protein sequentially around 60 aa each time through multiple steps and translocates rapidly. This contradicts our knowledge of protein unfolding, which is generally cooperative, particularly for titinI27, which is reported to unfold cooperatively or utmost through one intermediate during enzymatic unfolding by ClpX and ClpA.

We do not think this represents a contradiction. In fact, our observations are in good agreement with mechanical unfolding of tandem repeats of Titin I27 using AFM experiments (Oberhauser, Hansma, Carrion-Vazquez and Fernandez 2001). They showed that tandem repeats of TitinI27 unfolded in steps of ~22 nm. Dividing 22 nm by 0.34 nm/Amino Acid gives ~65 amino acids per unfolding event. This implies that, under force, ~65 amino acids of folded structure unfolds in a single step. This number is in excellent agreement with our kinetic step-size of 65 AA/step.

Importantly, the experiments cited by the reviewer on ClpA and ClpX are actually with ClpAP and ClpXP. We assert that this is an important distinction as we have shown that ClpA employs a different mechanism than ClpAP (Rajendar and Lucius 2010, Miller, Lin, Li and Lucius 2013, Miller and Lucius 2014). Thus, ClpA and ClpAP should be treated as different enzymes but, without question, ClpB and ClpA are different enzymes.

(2) It is also important to note that the unfolding of titinI27 from the N-terminus (as done in this study) has been reported to be very fast and cannot be the rate-limiting step as reported earlier(Olivares et al, PNAS, 2017). This contradicts the current model where unfolding is the rate-limiting step, and the translocation is assumed to be many orders faster than unfolding.

Most importantly, the Olivares paper is examining ClpXP and ClpAP catalyzed protein unfolding and translocation and not ClpB. These are different enzymes. Additionally, we have shown that ClpAP and ClpA translocate unfolded polypeptides with different rates, rate constants, and kinetic step-sizes indicating that ClpP allosterically impacts the mechanism employed by ClpA to the extent that even ClpA and ClpAP should be considered different enzymes (Rajendar and Lucius 2010, Miller, Lin, Li and Lucius 2013). We would further assert that there is no reason to assume ClpAP and ClpXP would catalyze protein unfolding using the same mechanism as ClpB as we do not think it should be assumed ClpA and ClpX use the same mechanism as ClpAP and ClpXP, respectively.

The Olivares et al paper reports a dwell time preceding protein unfolding of ~0.9 and ~0.8 s for ClpXP and ClpAP, respectively. The inverse of this can be taken as the rate constant for protein unfolding and would yield a rate constant of ~1.2 s-1, which is in good agreement with our observed rate constant of 0.9 – 4.3 s-1 depending on the ATP:ATPγS mixing ratio. For ClpB, we propose that the slow unfolding is then followed by rapid translocation on the unfolded chain where translocation by ClpB must be much faster than for ClpAP and ClpXP. We think this is a reasonable interpretation of our results and not a contradiction of the results in Olivares et al. Moreover, this is completely consistent with the mechanistic differences that we have reported, using the same single-turnover stopped flow approach on the same unfolded polypeptide chains with ClpB, ClpA, and ClpAP (Rajendar and Lucius 2010, Miller, Lin, Li and Lucius 2013, Miller and Lucius 2014, Li, Weaver, Lin et al. 2015).

(3) The model assumes the same time constant for all the unfolding steps irrespective of the secondary structural interactions.

Yes, we contend that this is a good assumption because it represents repetition of protein unfolding catalyzed by ClpB upon encountering the same repeating structural elements, i.e. Beta sheets.

(4) Unlike other single-molecule optical tweezer-based assays, the study cannot distinguish the unfolding and translocation events and assumes that unfolding is the rate-limiting step.

Although we cannot, directly, distinguish between protein unfolding and translocation we have logically concluded that protein unfolding is likely rate limiting. This is because the large kinetic step-size represents the collapse of ~60 amino acids of structure between two rate-limiting steps, which we interpret to represent cooperative protein unfolding induced by ClpB. It is not an assumption it is our current best interpretation of the observations that we are now seeking to further test.

**Reviewer #3 (Public Review):**
Summary:The authors have devised an elegant stopped-flow fluorescence approach to probe the mechanism of action of the Hsp100 protein unfoldase ClpB on an unfolded substrate (RepA) coupled to 1-3 repeats of a folded titin domain. They provide useful new insight into the kinetics of ClpB action. The results support their conclusions for the model setup used.Strengths:The stopped-flow fluorescence method with a variable delay after mixing the reactants is informative, as is the use of variable numbers of folded domains to probe the unfolding steps.Weaknesses:The setup does not reflect the physiological setting for ClpB action. A mixture of ATP and ATPgammaS is used to activate ClpB without the need for its co-chaperones, Hsp70. Hsp40 and an Hsp70 nucleotide exchange factor. This nucleotide strategy was discovered by Doyle et al (2007) but the mechanism of action is not fully understood. Other authors have used different approaches. As mentioned by the authors, Weibezahn et al used a construct coupled to the ClpA protease to demonstrate translocation. Avellaneda et al used a mutant (Y503D) in the coiled-coil regulatory domain to bypass the Hsp70 system. These differences complicate comparisons of rates and step sizes with previous work. It is unclear which results, if any, reflect the in vivo action of ClpB on the disassembly of aggregates.

We agree with the reviewer, there are several strategies that have been employed to bypass the need for Hsp70/40 or KJE to simplify in vitro experiments. Here we have developed a first of its kind transient state kinetics approach that can be used to examine processive protein unfolding. We now seek to go forward with examining the mechanisms of hyperactive mutants, like Y503D, and add the co-chaperones so that we can address the limitations articulated by the reviewer. In fact we already began adding DnaK to the reaction and found that DnaK induced ClpB to release the polypeptide chain (Durie, Duran and Lucius 2018). However, the sequential mixing strategy developed here was needed to go forward with examining the impact of co-chaperones.

**Recommendations for the authors:**

**Reviewer #1 (Recommendations For The Authors):**
Line 1: I recommend changing the title of the paper to remove the terms that are not clearly defined in the text: "robust" and "processive". What are the Authors' criteria for describing a molecular machine as "robust" vs. "not robust"? A definition of processivity is given in equation 2, but its value for ClpB is not reported in the text, and the criteria for classifying a machine as "processive" vs. "non-processive" are not included. Besides, the Authors have previously reported that ClpB is non-processive (Biochem. J., 2015), so it is now clear that a more nuanced terminology should be applied to this protein. Also, *Escherichia coli* should be fully spelled out in the title.

The title has been changed. We have removed “robust” as we agree with the reviewer, there is no way to quantify “robust”. However, we have kept “processive” and have added to the discussion a calculation of processivity since we can quantify processivity. Importantly, the unstructured substrates used in our previous studies represent translocation and not protein unfolding. here, on folded substrates, we detect rate-limiting protein unfolding followed by rapid translocation. Thus, we report a lower bound on protein unfolding processivity of 362 amino acids.

Line 20: The comment about mitochondrial SKD3 should be removed. SKD3, like ClpB, belongs to the AAA+ family, and it is simply a coincidence that the original study that discovered SKD3 termed it an Hsp100 homolog. The similarity between SKD3 and ClpB is limited to the AAA+ module, so there are many other metazoan ATPases, besides SKD3, that could be called homologs of ClpB, including mitochondrial ClpX, ER-localized torsins, p97, etc.

Removed.

Lines 133-139. Contrary to what the authors state, it is not clear that the "lag-phase" becomes significantly shorter for subsequent mixing experiments (Figure 1E) perhaps except for the last one (2070s). It is clear, however, that the emission enhancement becomes stronger for later mixes. This effect should be discussed and explained, as it suggests that the pre-equilibrations shorter than ~2000 sec do not produce saturation of ClpB binding to the substrate.

We have added supplemental figure 2, which represents a zoom into the lag region. This better illustrates what we were seeing but did not clearly show to the reader. In addition, we address all three changes in the time courses, i.e. extend of lag, change in peak position, and the change in peak height.

Line 175. The hydrolysis rate of ATPgammaS in the presence of ClpB should be measured and compared to the hydrolysis rate with ATP/ATPgammaS to check if the ratio of those rates agrees with the ratio of the translocation rates. These experiments should be performed with and without the RepA-titin substrate, which could reveal an important linkage between the ATPase engine and substrate translocation. These experiments are essential to support the claim of substrate translocation and unfolding with ATPgammaS as the sole energy source.

The time courses shown in figure 2 and supplemental Figure 1 are collected with only ATPgS and no ATP. The time courses show a clear increase in lag and appearance of a peak with increasing number of tandem repeats of titin domains. We do not see an alternate explanation for this observation other than ATPγS supports ClpB catalyzed protein unfolding and translocation. What is the reviewers alternate explanation for these observations?

We agree with the reviewer that the linkage of ATP hydrolysis to protein unfolding and translocation is essential and we are seeking to acquire this knowledge. However, a simple comparison of the ratio of rates is not adequate. We contend that a complete mechanistic study of ATP turnover by ClpB is required to properly address this linkage and such a study is too substantial to be included here but is currently underway.

All that said, the statement on line 175 was removed since we do not report any ATPase measurements in this paper.

Line 199: It is an over-simplification to state that "1:1 mix of ATP to ATPgammaS replaces the need for co-chaperones". This sentence should be corrected or removed. The ClpB co-chaperones (DnaK, DnaJ, GrpE) play a major role in targeting ClpB to its aggregated substrates in cells and in regulating the ClpB activity through interactions with its middle domain. ATPgammaS does not replace the co-chaperones; it is a chemical probe that modifies the mechanism of ClpB in a way that is not entirely understood.

We agree with the reviewer. The sentence has been modified to point out that the mix of ATP and ATPγS activates ClpB.

Figure 3B, Supplementary Figure 5A. The solid lines from the model fit cannot be distinguished from the data points. Please modify the figures' format to clearly show the fits and the data points.

Done.

Lines 326, 329. It is not clear why the authors mention a lack of covalent modification of substrates by ClpB. AAA+ ATPases do not produce covalent modifications of their substrates.

The issue of covalent modification was presented in the introduction lines 55 – 60 pointing out that much of what we have learned about protein unfolding and translocation catalyzed by ClpA and ClpX is from the observations of proteolytic degradation catalyzed by the associated protease ClpP. However, this approach is not possible for ClpB/Hsp104 as these motors do not associate with a protease unless they have been artificially engineered to do so.

Lines 396-399. I am puzzled why the authors try to correlate the size of the detected kinetic step with the length of the ClpB channel instead of the size characteristics of the substrate.

We are attempting to discuss/rationalize the observed large kinetic step-size which, in part, is defined by the structural properties of the enzyme as well as the size characteristics of the substrate. We have attempted to clarify this and better discuss the properties of the substrate as well as ClpB.

As I mentioned in the Public Review, it is essential to demonstrate that the emission increase used as the only readout of the ClpB position along the substrate is indeed caused by the proximity of ClpB to the fluorophore. One way to accomplish that would be to place the fluorophore upstream from the first I27 domain and determine if the "lag phase" in the emission enhancement disappears.

Alexa Fluor 555 is well established to exhibit PIFE. However, as in the response to the public review, we have included an appropriate control showing this in supplemental Fig. 1.

Finally, the authors repetitively place their results in opposition to the study of Weibezahn et al. published in 2004 which first demonstrated substrate translocation by engineering a peptidase-associated variant of ClpB. It should be noted that the field of protein disaggregases has moved since the time of that publication from the initial "from-start-to-end" translocation model to a more nuanced picture of partial translocation of polypeptide loops with possible substrate slipping through the ClpB channel and a dynamic assembly of ClpB hexamers with possible subunit exchange, all of which may affect the kinetics in a complex way. However, the present study confirmed the "start-to-end" translocation model, albeit for a non-physiological ClpB substrate, and that is the take-home message, which should be included in the text.

It is not clear to us that the field has “moved on” since Weibezahn et al 2004. Their engineered construct that they term “BAP” with ClpP is still used in the field despite us reporting that proteolytic degradation is observed in the absence of ATP with that system (Li, Weaver, Lin et al. 2015) and should, therefore, not be used to conclude processive energy driven translocation. The “partial translocation” by ClpB is also grounded in observations of partial degradation catalyzed by ClpP with BAP from the same group (Haslberger, Zdanowicz, Brand et al. 2008). It is not clear to us that the idea of subunit exchange leading to the possibility of assembly around internal sequences is being considered. We do agree that this is an important mechanistic possibility that needs further interrogation. We agree with the reviewer, all these factors are confounding and lead to a more nuanced view of the mechanism.

All that said, we have removed some of the opposition in the discussion.

**Reviewer #2 (Recommendations For The Authors):**
(1) It is assumed that the lag phase will be much longer than the phase in which we see a gradual increase in fluorescence, as the effect of PIFE is significant only when the enzyme is very close to the fluorophore. Particularly for RepA-titin3, the enzyme has to translocate many tens of nm before it is closer to the C-terminus fluorophore. However, in all cases, the lag time is lower or similar to the gradual increase phase (for example, Figure 3B). Could the authors explain this?

The extent of the lag, or time zero until the signal starts to increase, is interpreted to indicate the time the motor moves from it’s initial binding site until it gets close enough to the fluorophore that PIFE starts to occur. In our analysis we apply signal change to the last intermediate and dissociation or release of unfolded RepA-TitinX. The increase in PIFE is not “all or nothing”. Rather, it is starting to increase gradually. Further, because these are ensemble measurements, and each molecule will exhibit variability in rate there is increased breadth of the peak due to ensemble averaging.

(2) Although the reason for differences in the peak position (for example, Figure 1E, 2B) is apparent, the reason for variations in the relative intensities has to be given or speculated.

We have addressed the reason for the different peak heights in the revised manuscript. It is the consequence of the fact that each substrate has slightly different fluorescent labeling efficiencies. Thus, for each sample there is a mix of labeled and unlabeled substrates both of which will bind to ClpB but the unlabeled ClpB bound substrates do not contribute to the fluorescence signal, but will represent a binding competitor. Thus, for low labeling efficiency there is a lower concentration of ClpB bound to fluorescent RepA-Titinx and for higher labeling efficiency there is higher concentration of ClpB bound to RepA-Titinx leading to an increased peak height. RepA-Titin2 has the highest labeling efficiency and thus the largest peak height.

**Reviewer #3 (Recommendations For The Authors):**
The authors should make it clear that they and previous authors have used different constructs or conditions to bypass the physiological regulation of ClpB action by Hsp70 and its co-factors as mentioned above. In particular, the construct used by Avellaneda et al should be explained when they challenge the findings of those authors.Minor points:The lines fitting the experimental points are difficult or impossible to see in Figures 2B, 3B, and s5B.

Fixed

Typo bottom of p6 - "averge"

Fixed

Avellaneda, M. J., K. B. Franke, V. Sunderlikova, B. Bukau, A. Mogk and S. J. Tans (2020). "Processive extrusion of polypeptide loops by a Hsp100 disaggregase." Nature.

Doyle, S. M., J. Shorter, M. Zolkiewski, J. R. Hoskins, S. Lindquist and S. Wickner (2007). "Asymmetric deceleration of ClpB or Hsp104 ATPase activity unleashes protein-remodeling activity." Nature structural & molecular biology **14**(2): 114-122.

Durie, C. L., E. C. Duran and A. L. Lucius (2018). "*Escherichia coli* DnaK Allosterically Modulates ClpB between High- and Low-Peptide Affinity States." Biochemistry **57**(26): 3665-3675.

Haslberger, T., A. Zdanowicz, I. Brand, J. Kirstein, K. Turgay, A. Mogk and B. Bukau (2008). "Protein disaggregation by the AAA+ chaperone ClpB involves partial threading of looped polypeptide segments." Nat Struct Mol Biol **15**(6): 641-650.

Li, T., J. Lin and A. L. Lucius (2015). "Examination of polypeptide substrate specificity for *Escherichia coli* ClpB." Proteins **83**(1): 117-134.

Li, T., C. L. Weaver, J. Lin, E. C. Duran, J. M. Miller and A. L. Lucius (2015). "*Escherichia coli* ClpB is a non-processive polypeptide translocase." Biochem J **470**(1): 39-52.

Miller, J. M., J. Lin, T. Li and A. L. Lucius (2013). "*E. coli* ClpA Catalyzed Polypeptide Translocation is Allosterically Controlled by the Protease ClpP." Journal of Molecular Biology **425**(15): 2795-2812.

Miller, J. M. and A. L. Lucius (2014). "ATP-gamma-S Competes with ATP for Binding at Domain 1 but not Domain 2 during ClpA Catalyzed Polypeptide Translocation." Biophys Chem **185**: 58-69.

Oberhauser, A. F., P. K. Hansma, M. Carrion-Vazquez and J. M. Fernandez (2001). "Stepwise unfolding of titin under force-clamp atomic force microscopy." Proc Natl Acad Sci U S A **98**(2): 468-472.

Rajendar, B. and A. L. Lucius (2010). "Molecular mechanism of polypeptide translocation catalyzed by the *Escherichia coli* ClpA protein translocase." J Mol Biol **399**(5): 665-679.

Weibezahn, J., P. Tessarz, C. Schlieker, R. Zahn, Z. Maglica, S. Lee, H. Zentgraf, E. U. Weber-Ban, D. A. Dougan, F. T. Tsai, A. Mogk and B. Bukau (2004). "Thermotolerance requires refolding of aggregated proteins by substrate translocation through the central pore of ClpB." Cell **119**(5): 653-665.